
# Observations of sesquiterpenes and their oxidation products in central Amazonia during the wet and dry seasons

Lindsay D. Yee[1], Gabriel Isaacman-VanWertz[1*], Rebecca A. Wernis[2], Meng Meng[3**], Ventura Rivera[3], Nathan M. Kreisberg[4], Susanne V. Hering[4], Mads S. Bering[5], Marianne Glasius[5], Mary Alice Upshur[6], Ariana Gray Bé[6], Regan J. Thomson[6], Franz M. Geiger[6], John H. Offenberg[7], Michael Lewandowski[7], Ivan Kourtchev[8], Markus Kalberer[8], Suzane de Sá[9], Scot T. Martin[9,10], M. Lizabeth Alexander[11], Brett B. Palm[12], Weiwei Hu[12], Pedro Campuzano-Jost[12], Douglas A. Day[12], Jose L. Jimenez[12], Yingjun Liu[9***], Karena A. McKinney[9****], Paulo Artaxo[13], Juarez Viegas[14], Antonio Manzi[14], Maria B. Oliveira[15], Rodrigo de Souza[15], Luiz A. T. Machado[16], Karla Longo[17], and Allen H. Goldstein[1]

[1]Department of Environmental Science, Policy, and Management, University of California, Berkeley, Berkeley, 94720, USA
*Now at Department of Civil and Environmental Engineering, Virginia Tech, Blacksburg, 24061, USA
[2]Department of Civil and Environmental Engineering, University of California, Berkeley, Berkeley, 94720, USA
[3]Department of Chemical Engineering, University of California, Berkeley, Berkeley, 94720, USA
**Now at Department of Chemical Engineering and Applied Chemistry, University of Toronto, Toronto, CA
[4]Aerosol Dynamics Inc., Berkeley, 94710, USA
[5]Department of Chemistry, Aarhus University, 8000 Aarhus C, Denmark
[6]Department of Chemistry, Northwestern University, Evanston, 60208, USA
[7]National Exposure Research Laboratory, Exposure Methods and Measurements Division, United States Environmental Protection Agency, Research Triangle Park, 27711, USA
[8]Department of Chemistry, University of Cambridge, Cambridge, CB2 1EW, United Kingdom
[9]School of Engineering and Applied Sciences, Harvard University, Cambridge, 02138, USA
***Now at Department of Environmental Science, Policy, and Management, University of California, Berkeley, Berkeley,
****Now at Department of Chemistry, Colby College, Waterville, Maine, 04901, USA
[10]Department of Earth and Planetary Sciences, Harvard University, Cambridge, 02138, USA
94720, USA
[11]Environmental Molecular Sciences Laboratory, Pacific Northwest National Laboratory, Richland, 99352, USA
[12]Dept. of Chemistry and Cooperative Institute for Research in Environmental Sciences (CIRES), University of Colorado, Boulder, 80309, USA
[13]Department of Applied Physics, University of São Paulo, SP, Brazil
[14]Instituto Nacional de Pesquisas da Amazonia, Manaus, AM, Brazil
[15]Universidade do Estado do Amazonas, Manaus, AM, Brazil;
[16]Instituto Nacional de Pesquisas Espiacais, São José dos Campos, SP, Brazil
[17]Instituto Nacional de Pesquisas Espiacais, Cachoeira Paulista, SP, Brazil

*Correspondence to*: Lindsay D. Yee (lindsay.yee@berkeley.edu)

**Abstract.** Biogenic volatile organic compounds (BVOCs) from the Amazon forest region represent the largest source of organic carbon emissions to the atmosphere globally. These BVOC emissions dominantly consist of volatile and intermediate volatility terpenoid compounds that undergo chemical transformations in the atmosphere to form oxygenated condensable gases and secondary organic aerosol (SOA). We collected quartz filter samples with 12-hour time resolution and performed hourly in-situ measurements with the Semi-Volatile Thermal desorption Aerosol Gas chromatograph (SV-TAG) at a rural site ("T3") located to the west of the urban center of Manaus, Brazil as part of the Green Ocean Amazon (GoAmazon2014/5) field campaign to measure intermediate volatility and semi-volatile BVOCs and their oxidation products during the wet and dry seasons. We speciated and quantified 30 sesquiterpenes and four diterpenes with concentrations in the range 0.01–6.04 ng m$^{-3}$ (1–670 ppq$_v$). We estimate that sesquiterpenes contribute approximately 14% and 12% to the total reactive loss of $O_3$ via reaction with isoprene or terpenes during the wet and dry seasons, respectively. This is reduced from ~50-70% for within-canopy reactive $O_3$ loss, attributed to ozonolysis of highly reactive sesquiterpenes (e.g. β-caryophyllene) that are reacted away before reaching our measurement site. We further identify a suite of their oxidation products in the gas and particle phases and explore their role in biogenic SOA formation in the Central Amazon region. Synthesized authentic standards were also



used to quantify gas- and particle-phase oxidation products derived from β-caryophyllene. Using tracer-based scaling methods for these products, we roughly estimate that sesquiterpene oxidation contributes at least 1-18% (median 5%) of total submicron OA mass. However, this is likely a low-end estimate, as evidence for additional unaccounted sesquiterpenes and their oxidation products clearly exists. By comparing our field data to laboratory-based sesquiterpene oxidation experiments we confirm more

than 40 additional observed compounds produced through sesquiterpene oxidation are present in Amazonian SOA, warranting further efforts towards more complete quantification.

## 1 Introduction

Emission of biogenic volatile organic compounds (BVOCs) from terrestrial vegetation represents a large source of organic carbon to Earth's atmosphere. These emissions comprise a wide array of chemical species, including known

terpenoids: isoprene ($C_5$), monoterpenes ($C_{10}$), sesquiterpenes ($C_{15}$), and diterpenes ($C_{20}$). Isoprene, 2-methyl-1,3-butadiene ($C_5H_8$), is a hemi-terpene and the most dominant non-methane hydrocarbon emitted to the atmosphere at levels of ~500 TgC/year globally (Guenther et al., 2006, 2012). Global emissions of terpenes are estimated for monoterpenes ($C_{10}H_{16}$) at ~160 TgC/year, sesquiterpenes ($C_{15}H_{24}$) at ~30 Tg/year, and global emission estimates for diterpenes ($C_{20}H_{42}$) have yet to be made (Guenther et al., 2012).

In the atmosphere, such compounds undergo chemical transformations that lead to the formation of biogenic secondary organic aerosol (BSOA) and affect local radical budgets (Kesselmeier et al., 2013; Lelieveld et al., 2008, 2016) and carbon cycling (Bouvier-Brown et al., 2012; Guenther, 2002). Globally, the majority of organic aerosols stem from oxidation of biogenic carbon, yet their role in affecting Earth's radiative balance remains unclear (NAS, 2016). This is largely due to limited observations of the speciated precursors and identification of their oxidation products, which are crucial for

understanding their chemical pathways and fate in Earth's atmosphere (Worton et al., 2012).

While laboratory measurements simulating oxidation of BVOCs provide insight into atmospheric chemistry, challenges still exist for making ambient, high time-resolution, speciated measurements of the higher carbon number terpenes and their oxidation. These compounds present measurement challenges due to several reasons. First, their relatively low vapour pressure makes sample collection more challenging due to line losses. Second, they tend to be present at lower ambient

concentrations due to lower emission and higher reactivity, therefore requiring very sensitive detection methods and/or lower time-resolution. These measurement challenges have resulted in more research focused on highly volatile and higher concentration BVOCs (i.e. isoprene, monoterpenes), but there is less understanding of sesquiterpenes (Bouvier-Brown et al., 2009a, 2009b; Chan et al., 2016), and almost no data on concentrations or chemical reaction pathways of diterpenes.

Isoprene has a laboratory determined SOA yield of <6% (Dommen et al., 2006; King et al., 2010; Kroll et al., 2005,

2006; Xu et al., 2014) and an estimated 3.3% yield in the SE U.S. (Marais et al., 2016), but nevertheless contributes a major fraction of organic aerosol over forested regions because it is emitted in such high quantities relative to other BVOCs (Carlton et al., 2009; Hu et al., 2015). Though other terpenes are present in the atmosphere at much lower concentrations, they generally react faster with oxidants such as OH, $O_3$, and $NO_3$ (Atkinson, 1997), and have higher SOA yields at typical atmospheric OA concentrations, ~5-10% for monoterpenes as reported in Griffin et al., 1999a, 1999b and ~20-70% for sesquiterpenes as

reported in Chen et al., 2012; Griffin et al., 1999b; Hoffmann et al., 1997; Jaoui et al., 2013; Lee et al., 2006a, 2006b. SOA yields from diterpenes have not yet been quantified, though they are likely higher than those of sesquiterpenes due to their higher carbon number and lower volatility. While emitted mass generally decreases with decreasing volatility (and increasing carbon number), the concomitant rise in sheer possible number of compounds from $C_5$ to $C_{10}$ to $C_{15}$ to $C_{20}$ backbones and associated SOA yields and oxidant reactivity indicates that lower-volatility terpenes may have an important impact on regional

chemistry and BSOA formation.



Using the BVOC emission model, MEGAN2.1, it is predicted that ~80% of terpenoid emissions come from tropical trees that cover about 20% of the global land surface (Guenther et al., 2012), yet very few observations of sesquiterpenes exist from these regions. Chemical characterization of tropical plant tissue shows the presence of an abundance of sesquiterpenes (Chen et al., 2009a; Courtois et al., 2012) and suggests their widespread emission from such vegetation (Chen et al., 2009a; Courtois et al., 2009). Previous branch enclosure measurements of native Amazon saplings found that sesquiterpene emission was below detection limits (Bracho-Nunez et al., 2013) even though sesquiterpenes have been observed within and outside the canopy of the Central Amazon (Alves et al., 2016; Jardine et al., 2012). Further, Chen et al., 2009a observed higher sesquiterpene emissions from wounded seedlings of the tropical tree, *Copaifera officinalis*, with similar composition and quantity to that in the leaves. However, while ambient concentrations of sesquiterpenes have been measured by field-deployable mass spectrometers such as Proton Transfer Time-of-Flight Mass Spectrometry (Jordan et al., 2009), previous measurements have provided little or no separation of isomers, which can vary substantially (orders of magnitude) in their reactivity and SOA yields, so a significant knowledge gap remains regarding the contribution of these compounds to organic aerosol.

Aerosols play a critical role in cloud formation and the hydrologic cycle of the Brazilian Amazon (Fan et al., 2018; Pöschl et al., 2010; Wang et al., 2016), which is also one of the major source regions of global BVOCs. Previous studies have found the aerosols over this region (Martin et al., 2010) to be primarily composed of organic material (Artaxo et al., 2013) derived from BVOCs (Chen et al., 2009b, 2015). While isoprene oxidation has been estimated to contribute ~50% of OA in Amazonia (Chen et al., 2015), of which a large portion is attributed to uptake of isoprene-epoxydiols (Hu et al., 2015; de Sá et al., 2017), the remaining contribution to OA from other BVOC precursors such as monoterpenes and sesquiterpenes remains largely unconstrained. Khan et al., 2017 report that including updated sesquiterpene emissions and SOA pathways (all represented by β-caryophyllene mechanism) in the STOCHEM global chemical transport model led to a 48% increase in global SOA burden and 57% increase in SOA production rate. In addition, the highest concentrations of sesquiterpene-derived SOA were modelled to be present over central Africa and South America. However, the general lack of available time-resolved measurements of speciated sesquiterpenes and their oxidation products in either the gas or particle phase has precluded fully constraining their contribution to BSOA formation. For terpenes in general there are even greater measurement challenges associated with observation of tracers of their oxidation: the gas-phase component of these semi-volatile compounds may condense in sample lines or be lost by wet/dry deposition, or quickly react to form compounds sufficiently ubiquitous to be disconnected from a specific precursor. Particle-phase composition is therefore critical for studying the importance of individual terpenes and terpene classes, but dynamic gas-particle partitioning of these semi-volatile products requires the contemporaneous measurement of the difficult-to-measure gas-phase components. Measurements of gas/particle partitioning and concentrations of biogenic OA tracers from isoprene and monoterpene oxidation have been reported previously as part of the Green Ocean Amazon (GoAmazon2014/5) field campaign (Isaacman-VanWertz et al., 2016), but few, if any, sesquiterpene oxidation products were identified in the dataset at that time. To fully characterize the sources of OA in the region, molecular-level and chemically specific signatures of oxidation products from a more complete range of BVOC precursors need to be identified and quantified.

Here we report the first speciated measurement of 30 sesquiterpenes and four diterpenes in the Central Amazon and assess their role in reactive oxidant losses during the wet and dry seasons through in-situ observations with the Semi-Volatile Thermal desorption Aerosol Gas chromatograph (SV-TAG) and off-line measurements of quartz filter collected aerosol samples. We further report measurements of specific oxidation products of the sesquiterpene β-caryophyllene that were synthesized in the laboratory and attribute more than 40 additional species observed on a representative filter sample from the wet season to sesquiterpene oxidation based on comparison to products found in laboratory-based oxidation experiments. Finally, we provide a rough low-end estimate of the contribution of sesquiterpene oxidation to OA in the region.





## 2 Experimental section

### 2.1 Green Ocean Amazon (GoAmazon2014/5) field campaign: "T3" site description

Measurements were conducted as part of the Green Ocean Amazon (GoAmazon2014/5) field campaign, for which the scientific objectives and measurement sites are described in an overview paper by Martin et al., 2016. We present
observations from the wet and dry seasons of 2014, respectively referred to as Intensive Operating Periods 1 and 2 (IOP1 and IOP2). IOP1 measurements were taken 1 February to 31 March 2014 and IOP2 measurements were taken 15 August to 15 October 2014 at the "T3" Manacapuru rural site, located 70 km downwind of urban Manaus. The T3 site was located on a cleared pasture site (−3.2133°, −60.5987°) and 2 km north of the nearest heavily travelled road which connects Manaus to Manacapuru. The immediate surrounding forest (~2.5 km away) consisted primarily of secondary forest, and prevailing
meteorological conditions (northeasterly winds) resulted in regional transport of clean air from the primary forest north and east of the site mixed with the outflow of Manaus pollution from the east.

    Several instruments for investigating gas and particle phase chemistry were housed at T3 alongside the instrument suite of the United States Department of Energy Atmospheric Radiation Measurement (ARM) Climate Research Facility. ARM Trailers consisted of the Atmospheric Radiation Measurement Facility One (AMF-1) and Mobile Aerosol Observing
System (MAOS) (Mather and Voyles, 2013). Here we focus on measurements conducted using a Semi-Volatile Thermal desorption Aerosol Gas Chromatograph (SV-TAG) and quartz filters collected using a custom-designed sequential filter sampler. Additional details of these and supporting measurements follow.

### 2.2 Deployment of a Semi-Volatile Thermal desorption Aerosol Gas chromatograph (SV-TAG)

We deployed a Semi-Volatile Thermal desorption Aerosol Gas chromatograph (SV-TAG) housed in one of the
instrument trailers at T3. Details of the instrument development and operation have been published previously (Isaacman-VanWertz et al., 2016; Isaacman et al., 2014; Kreisberg et al., 2009; Williams et al., 2006; Zhao et al., 2013) and we describe SV-TAG deployment in brief here. During the campaign, ambient air was pulled through a 15.24 cm I.D. duct at ~5 m above ground level. Flow through the ducting was maintained at laminar flow conditions with minimal residence time to minimize loss of semi-volatile species from the center-line of flow. Air was then sub-sampled from the center of the sampling duct at
20 lpm for 22 min through a cyclone (PM$_1$ cutpoint) to SV-TAG's dual collection cells held at 32 °C. Concentrations and gas-particle partitioning were measured through a combination of particle-only measurements, in which gas-phase components were removed through a multi-channel carbon monolith denuder (500 channels, 30 mm OD x 40.6 cm; MAST Carbon) upstream of one collection cell, and total gas-plus-particle measurements, with no removal of the gas-phase component. One collection cell always sampled total gas-plus-particle concentrations, while the other cell alternated between particle-only
samples and total gas-plus-particle samples, which were used to remove any bias between collection cells. Particle fraction, $F_p$, is calculated comparing the particle-only signal from the denuded channel to the gas and particle signal from the other cell. Measurements of $F_p$ for many tracers of biogenic origin from this campaign were presented previously in Isaacman-VanWertz et al., 2016 and an intercomparison of SV-TAG and other instrumentation for measuring $F_p$ is described in Thompson et al., 2017. We found here that sesquiterpenes and their oxidation products detected by SV-TAG were completely in the gas phase
($F_p$ was zero, as discussed in Sections 3.1 Chemical characterization of observed sesquiterpenes and 3.2.3 Observation of β-caryophyllene oxidation products), so we do not present time-dependent $F_p$ measurements for these compounds.

    After sample collection, material was thermally desorbed from the collection cells into helium (He) at a rate of 35° C min$^{-1}$ up to 320 ° C, taking approximately 8 mins. The helium was saturated with a derivatization agent N-methyl-N-(trimethylsilyl) trifluoroacetamide ("MSTFA"), which converts polar –OH moieties to –OSi(CH$_3$)$_3$ (trimethylsilyl ester)
groups for separation on a gas chromatography column (Isaacman et al., 2014). Desorbed material was focused onto a pre-concentrator held at 30 °C, and then transferred onto a gas chromatography column (Restek, Rtx-5Sil MS, 20 m × 0.18 mm ×



0.18 μm) via a valveless injector (Kreisberg et al., 2014). Analysis was performed using an Agilent 7890A/5975 Gas
Chromatograph (GC) coupled to a quadrupole mass spectrometer (MS). The GC program consisted of He flow at 1 sccm and
a ramp from 50 °C to 330 °C (ramp rate 23.6 °C min⁻¹), then holding at 330 °C for 2.2 min while He flow ramped to 3 sccm.
The use of online-derivatization greatly extends the use of SV-TAG for analysis of highly oxygenated species typical of BVOC

oxidation, but it also renders more complex chromatograms. For many of the observed sesquiterpenes, their individual
contributions to the total ion signal within a chromatogram were low. Thus, analysis was occasionally performed (~every 13
hrs) without derivatization to allow for clearer identification and quantification of all detected sesquiterpene species.

### 2.2.1 Compound identification

A typical sample total ion chromatogram (TIC) contains hundreds to thousands of compounds. For peak
deconvolution, compound peaks are separated and quantified using a characteristic ("selected") ion fragment. Chemical
identification is aided by comparing the peak's background-subtracted mass spectrum and $n$-alkane based retention index to
those of authentic standards run on SV-TAG or presented in the literature and available mass spectral libraries. The 2014
NIST/EPA/NIH Mass Spectral Library (Stein et al., 2014), the Adams essential oil library (Adams, 2007), and a proprietary
library from a flavour and fragrance company (MANE) were used for mass spectral matching. Spectral matches are considered
perfect if the match statistic (Stein, 1994) is 999, excellent if >900, good if 800–900, and fair if 700–800. Further, additional
work using EI mass spectral matching to identify components of a complex OA sample found that the probability of incorrect
identification was low (30% and 14%) for match statistics between 800–900 and >900, respectively (Worton et al., 2017).
Here we present proposed identities if the spectral match factor is >800 and the retention index is reasonable with the proposed
identity. Retention index ($RI$) is helpful for determining elution order of compounds and narrowing the possible compound
identities for species such as the sesquiterpenes with similar mass spectra. We calculate $n$-alkane based retention index for
compound $i$, using Eq. (1) below

$$RI = 100 \times \left[ n + \frac{t_i - t_n}{t_{n+1} - t_i} \right] \tag{1}$$

where $n$ is the number of carbon atoms of the $n$-alkane that elutes before species $i$, and $t$ represents retention time. While many
sesquiterpenes observed in ambient chromatograms are not available as authentic standards, sesquiterpene-rich essential oils
from Amazonian trees were injected on SV-TAG for chromatographic separation and to aid in identification. These included
Copaiba essential oil and Andiroba essential oil obtained from a local pharmacy in Manacapuru, Brazil, and additional bottles
of Copaiba essential oil from Young Living Essential Oils, Lehi, UT, and through personal communication (origin, Bolivia).

### 2.2.2 Compound quantification

Within a selected ion chromatogram (SIC), peak signal is integrated at the quantification ion and calibrated based on
the best available authentic standard. Peak-integrated ion signal of both the analyte and the standard is normalized by the
integrated ion signal of an isotopically-labelled internal standard in each sample to account for differences in recovery by
compound functionality in SV-TAG as well as changing MS detector response over time. For the sesquiterpenes, n-tetradecane
d₃₀ (CAS # 204244-81-5) was selected as the internal standard to normalize by. Reported oxidation products are normalized
using 2-C¹³-pentaerythritol. Compounds were quantified using authentic standards whenever possible, though analytical grade
standards for many of the observed sesquiterpenes are not commercially available and/or were not present in the custom
standard solution that was used for in-field calibrations. Longifolene, β-caryophyllene, and alloaromadendrene were the only
sesquiterpenes present in the calibration solution during deployment, and only β-caryophyllene and alloaromadendrene were
occasionally detected in ambient air. Post-deployment calibrations were performed in the laboratory with newly acquired
sesquiterpene and sesquiterpene-derived oxidation product standards using relative response factors to β-caryophyllene. An
average response factor for several sesquiterpenes was used to quantify compounds for which authentic standards were not




available (Chan et al., 2016). All sesquiterpenes are quantified on $m/z$ = 161; average sensitivity of most sesquiterpenes to quantification on this ion is 6.4 ± 6.0 times more sensitive than for β-caryophyllene.

Standards of several oxidation products from β-caryophyllene ozonolysis were custom-synthesized at Northwestern University. The oxidation products synthesized were β-caryophyllene aldehyde, β-nocaryophyllone aldehyde, β-

caryophyllonic acid, β-nocaryophyllonic acid, β-caryophyllinic acid, and β-nocaryophyllinic acid. Synthesis details and procedures are outlined in (Be et al., 2017). Each standard was analysed in the laboratory by SV-TAG and by two-dimensional gas chromatography with high-resolution-time-of-flight mass spectrometry (GC×GC-HR-ToFMS) to obtain MS and retention time information to aid identification of these compounds in SV-TAG chromatograms. Relative response factors of the synthesized standards to pinonic acid, a compound in the regular standard solution for SV-TAG during deployment, were

obtained and used for quantification.

### 2.2.3 Laboratory-generated SOA from sesquiterpenes

Filter samples from laboratory oxidation experiments of several sesquiterpenes were analysed by GC×GC-HR-ToFMS to provide mass spectral information for identification of potential sesquiterpene-derived oxidation products in ambient samples. Mass spectra of resolved peaks from these and previously analysed filters were added to custom MS

libraries and are listed in Table S4. Sesquiterpenes were oxidized in the U.S. EPA National Exposure Research Laboratory reactors (Table S5) in the dark via ozonolysis; some were also oxidized under conditions of OH oxidation in the presence of $NO_x$ according to methods described previously (Jaoui et al., 2003, 2004, 2013, 2016; Offenberg et al., 2017). The following sesquiterpene systems were studied: β-caryophyllene, α-cedrene, α-copaene, aromadendrene, β-farnasene, and α-humulene. In addition, a complex mixture rich in sesquiterpenes (copaiba essential oil, Amazon origin) was also oxidized

under ozonolysis conditions as a representation of the potential mixture expected in the Amazon atmosphere.

### 2.3 Supporting measurements

Several supporting measurements were made that allow for interpretation of the chemistry observed. These include gas-phase measurements of BVOCs from a Proton Transfer Time-of-Flight Mass Spectrometer (PTR-ToF-MS, Ionicon Analytik) and particle-phase measurements from an Aerodyne High Resolution Aerosol Mass Spectrometer (HR-ToFAMS),

hereinafter referred to as the AMS (DeCarlo et al., 2006). Operation and analysis procedures are outlined elsewhere for the PTR-ToF-MS (Liu et al., 2016) and for the AMS (de Sá et al., 2017). Positive matrix factorization analysis of AMS data was performed to resolve the statistical factor, IEPOX-SOA, which is considered to be a tracer for organic aerosol formed through particle uptake of isoprene epoxydiols and has been previously described (de Sá et al., 2017). Filter-based measurements were also taken using a custom-built sequential filter sampler. Selected filter samples were analysed using various chromatographic,

ionization, and mass spectrometric techniques to provide additional chemical insight. The sequential filter sampler and filter analysis techniques are described in brief in the following sections. Routine meteorology data and gas-phase measurements (e.g. $O_3$) were provided by the Atmospheric Radiation Measurement (ARM) Climate Research Facility, a U.S. Department of Energy Office of Science user facility sponsored by the Office of Biological and Environmental Research.

### 2.3.1 Sequential filter sampler

Aerosol samples were collected on quartz-fiber filters using a custom-built sequential filter sampler (Aerosol Dynamics, Inc.). Ambient air was sampled at 120 LPM through a 2.4 cm I.D. stainless steel tube 4 m above ground level. Sampled air passed through 2.7 m of 2 cm I.D. copper tubing kept at temperatures below the dew point of the trailer temperature for trapping excess water with periodic manual removal. The sample inlet geometry and flow conditions minimized particle losses (< 5%) for those between 10 and 1000 nm. We estimate 70% removal of intermediate volatility organic compounds

(IVOCs), minimizing adsorption of gas-phase organics onto the filters. Previous filter measurements have noted sampling



artefacts due to $O_3$ penetration (Dzepina et al., 2007). We estimate 90% removal of $O_3$ (estimated diffusive losses) for the sampler design used in this study, which should minimize further reaction of organics collected on filters.

Following the water removal stage was a pair of greaseless cyclones operating in parallel with aerodynamic diameter cutpoints of 1 μm. The cyclones are equivalent to the AIHL cyclone (John and Reischl, 1980) originally designed for $PM_{2.5}$

collection at 21.7 LPM but experimentally verified to provide $PM_1$ separation at 60 LPM. The sample was then introduced into a 91 cm length of 32 mm I.D. aluminium tubing, to one of six filter housings (HiQ, ILPH-102) containing a 101.6 mm diameter quartz fiber filter (Whatman, QM-A Quartz). Filter housings were modified from the manufacturer to remove all adhesives to prevent potential off-gas and contamination of the filter samples. Further modifications to the housings included replacing filter supports with custom etched 316L stainless steel support screens and utilizing an o-ring face seal not in contact

with the sample flow or filter.

Before deployment filters were pre-treated by baking at 550°C for 12 hours. During IOP1 (wet season), samples were collected at approximately 12-hour time resolution, from 06:15-18:00 and 18:30-06:15, local time (LT) basis. Filter changes occurred daily from 18:00-18:30 LT. Field blanks were also collected weekly in each filter holder. Filter samples collected during IOP2 (dry season) are not presented in this analysis. Removed filters were kept frozen (or transported on ice) until

analysis. Switching valves were automated to time sample collection appropriately over each filter throughout the day. Flow rates were logged in LabView using a TSI mass flow meter (TSI, model 4045).

**2.3.2 Two-dimensional gas chromatography with high-resolution-time-of-flight mass spectrometry (GC×GC-HR-ToFMS)**

Selected filter samples and standards were analysed in the laboratory using two-dimensional gas chromatography

with high-resolution-time-of-flight mass spectrometry. Aliquots of samples (multiples of ovoid filter punches with an area of 0.4 cm² each) were introduced into the gas chromatograph using a thermal-desorption autosampler (Gerstel, TDS-3 and TDSA2) with built-in derivatization using MSTFA. Compounds were separated first on a nonpolar column (Restek, Rxi-5Sil-MS, 60 m × 0.25 mm ×250 μm), then transferred to a Zoex Corporation cryogenic dual-stage thermal modulator comprised of guard column (Restek, 1.5 m × 0.25 mm, Siltek). The modulation period was 2.3 s, followed by separation on a secondary

column (Restek, Rtx-200MS, 1 m × 0.25 mm ×250 μm) to separate polar compounds. The GC temperature program ramped from 40 °C to 320 °C at a rate of 3.5 °C min⁻¹, holding for 5 min, and the He carrier gas flow was 2 mL min⁻¹. Following chromatographic separation, analysis was performed with a Tofwerk high-resolution $\left(\frac{m}{\Delta m} \approx 4000\right)$ mass spectrometer employing electron impact (70 eV) ionization.

To correct for compound transmission efficiency through the system, total ion signal for each peak was corrected

based on a volatility curve comprising even-carbon numbered perdeuterated alkanes as internal standards (from $C_{12}D_{26}$ through $C_{36}D_{74}$). The custom-synthesized β-caryophyllene oxidation products (described in Section 2.2.2 Compound quantification) and several filter samples containing sesquiterpene-derived SOA from laboratory oxidation experiments (described in Section 2.2.3 Laboratory-generated SOA from sesquiterpenes) were analysed to create characteristic mass spectra of sesquiterpene-derived oxidation products. These mass spectra were put into custom MS libraries and added to the available NIST 14 MS

libraries within NIST MS Search v.2.2 software to be searched against when analysing sample chromatograms. Custom MS libraries comprising mass spectra of products from previous laboratory oxidation experiments of the monoterpenes α-pinene, myrcene, and d-limonene were also included in MS searches. These custom MS libraries will be made available online for the use of the atmospheric chemistry community and are the subject of a future publication. Published MS in the literature were also used to identify previously reported tracers of isoprene and terpene oxidation as described in Section 3.3.2 Additional

identification of terpene oxidation products by mass spectral matching. For peaks with "good" MS match factors > 800 out of 1000 (Stein, 1994) and an alkane-based retention index matching within ± 10 of that of the library entry, a tentative match was



considered at least for source categorization of the peak. Some peaks with <800 MS match factors were included after manual review accounting for co-elutions and other factors affecting MS quality and MS matching.

### 2.3.3 Ultra-High Performance Liquid Chromatography Mass Spectrometry (UHPLC-MS)

A subset of filters were extracted and analysed using Ultra-High Performance Liquid Chromatography coupled to quadrupole time-of-flight MS (UHPLC-qTOF-MS) for carboxylic acids and organosulfates. The method was based on Kristensen and Glasius, 2011 and Kristensen et al., 2016. Monoterpene oxidation products were analysed using these methods and further optimized for analysis of sesquiterpene products. Filters were extracted in 1:1 methanol:acetonitrile assisted by sonication, evaporated to dryness using nitrogen gas, and reconstituted in 200 μL MilliQ water with 10% acetonitrile and 0.1% acetic acid. The UHPLC system used an Acquity T3 column (1.8 μm, 2.1 × 100 mm, Waters) with a mobile phase of eluent A: 0.1% acetic acid in MilliQ water and eluent B: acetonitrile with 0.1% acetic acid (eluent flow was 0.3 mL min$^{-1}$). The 18 min gradient was: Eluent B increased from 3% to 80% from 1 min to 12 min, and then increased to 100% (during 0.5 min) where it was held for 3 minutes, before returning to initial conditions. The qTOF-MS had an electrospray ionization source and was operated in negative ionization mode with a nebulizer pressure of 3.0 bar, dry gas flow 7.0 L min$^{-1}$, source voltage 3.0 kV, and transfer time of 50 μs. Oxidation products of β-caryophyllene were identified by comparison of their mass spectra with previous work (Alfarra et al., 2012; Chan et al., 2011; van Eijck et al., 2013; Jaoui et al., 2003) as well as products obtained from a smog chamber study of ozonolysis of β-caryophyllene. Quantification of β-caryophyllinic and β-nocaryophyllonic acid was performed using a synthesized standard provided by Dr. J. Parshintsev, Helsinki University following previously reported synthesis procedures (Parshintsev et al., 2010).

### 2.3.4 Nanoelectrospray (nanoESI) Ultra-High Resolution Mass Spectrometry (UHRMS)

Selected filter samples from the wet season were extracted and analysed by direct infusion ultra-high-resolution LTQ Orbitrap Velos mass spectrometer (Thermo Fisher, Bremen, Germany) equipped with a TriVersa Nanomate robotic nanoflow chip-based ESI (nanoESI, Advion Biosciences, Ithaca NY, USA) source according to methods described previously (Kourtchev et al., 2013, 2014, 2015, 2016). This analysis provided mass resolution power of ≥ 100 000 and mass accuracy <1 ppm to provide molecular assignments. The direct infusion nanoESI parameters were as follows: the ionization voltage and back pressure were set at −1.4 kV and 0.8 psi, respectively. The inlet temperature was 200 °C. The sample flow rate was approximately 200-300 nL min$^{-1}$. The negative ionization mass spectra were collected in three replicates over ranges $m/z$ 100–650 and $m/z$ 150–900 and processed using Xcalibur 2.1 software (Thermo Scientific). Chemical formulae of the form $C_cH_hN_nO_oS_s$ were made according to analysis procedures presented in Kourtchev et al., 2013, 2015; Zielinski et al., 2018 and only ions that were observed in all three replicate extract analyses were kept for evaluation. All sample intensities were normalized to the aerosol organic carbon loading as well.

## 3 Results

### 3.1 Chemical characterization of observed sesquiterpenes

Thirty sesquiterpene species were observed regularly in the gas phase in SV-TAG chromatograms during the GoAmazon campaign at T3. Sesquiterpenes with chemical formula C15H24 were mostly resolved within single ion chromatograms (Figure 1) by a characteristic ion, m/z 161, C12H17+, typically coincident with the molecular ion, m/z 204. In addition, a few sesquiterpenes with chemical formula C15H22 were resolved at their molecular ion (m/z 202), as well as a few diterpenes with chemical formula C20H32, resolved using characteristic ion, m/z 257. Compound names for those compounds positively identified via MS matching and retention index are labelled accordingly in chromatograms and listed



with mean concentrations observed during the wet and dry seasons in Table 1. In some cases, peak signal was too low to provide good MS matching, so retention index (RI) information was also used to propose identities.

While many sesquiterpenes observed in ambient chromatograms are not available as authentic standards, sesquiterpene-rich Copaiba and Andiroba essential oils from Amazonian trees were injected on SV-TAG for chromatographic

separation and to aid in identification. Copaiba essential oil originates from trees of the genus *Copaifera*, comprising over 70 species (Plowden, 2003), several of which are distributed throughout the greater Amazon region (SpeciesLink, 2018) While copaiba trees are commonly referred to as the "diesel" or "kerosene" tree due to the oil's limited use as a biofuel, the essential oil is extracted primarily for medicinal purposes. Andiroba essential oil is derived from *Carapa guianensis*, also widely used for medicinal purposes.

While the exact composition, grade, and quality of the essential oils depends on multiple factors (e.g. extraction method, species, origin), we hypothesized that the ambient sesquiterpene composition might reflect similar chemical composition to the essential oils of tree species prevalent in the area. Studies have previously observed the chemical similarity between that of terpene emissions and within plant content (Ormeno et al., 2010), and specifically so for *Copaifera officinalis* (Chen et al., 2009a). Most of the sesquiterpenes observed in ambient samples (13 Feb 2014 06:47 UTC (02:47 LT)) coincide

with those observed in copaiba essential oil (Figure 1, black and red traces respectively). This similarity between the ambient sesquiterpenes composition and essential oil composition also allowed for positive identification of some of the peaks in the ambient chromatogram. Sesquiterpene standards that were commonly available and brought to the field as standards included (+)-longifolene, β-caryophyllene, and alloaromadendrene with their chromatographic retention times indicated in Figure 1. Further, the essential oils injected on SV-TAG have relatively similar sesquiterpene composition (Table S1) and are

comparable to previously analysed essential oils/tissue from Amazonian trees (Table S2).

**3.2 Sesquiterpene oxidation**

We further explore in this section indicators for sesquiterpene oxidation in the region. First, by comparing relative concentrations of isoprene, monoterpenes, and sesquiterpenes at T3 compared to those made within the canopy at an upwind Amazon forest site (from previous literature), we confirm that the majority of sesquiterpene oxidation must occur within/near

the canopy. Second, differences in chemical composition between ambient samples at our site away from the canopy and that of sesquiterpene-rich essential oils (used as a proxy for emission profile within the canopy) reveals that the most reactive sesquiterpenes in the oils (e.g. β-caryophyllene) are reacted away to levels below detection by SV-TAG before reaching the sampling location at T3. Third, we observe known oxidation products of β-caryophyllene in SV-TAG and on filter samples collected at T3.

**3.2.1 Sesquiterpene contribution to total O₃ reactivity**

Sesquiterpenes are observed at concentrations much lower than those of the monoterpenes and isoprene (Figure Figure 2a and Figure 3a). They generally react relatively quickly with $O_3$, which dominates their reactive loss, and thus a strong anti-correlation with $O_3$ concentration is observed (Figure 2a and Figure 3a). During daytime, the levels of photochemically produced $O_3$ keep sesquiterpenes concentrations low. In addition, downward transport of ozone-rich air

during convective storms in the Amazon that typically occur during late morning or early afternoon hours (Gerken et al., 2016) also contribute to the temporal concentration profile observed with highest concentrations at night (Figure 4). This is in contrast with typically observed daytime maxima for isoprene and monoterpenes (Alves et al., 2016; Yáñez-Serrano et al., 2015), for which their reactive loss is dominated by reaction with OH and with OH/$O_3$, respectively (Atkinson, 1997; Kesselmeier et al., 2013; Kesselmeier and Staudt, 1999), and whose daytime emissions are likely more associated with

immediate release following production as a function of solar radiation input (Alves et al., 2014; Bracho-Nunez et al., 2013; Harley et al., 2004; Jardine et al., 2015; Kuhn, 2002; Kuhn et al., 2004).



To better understand reactive loss of $O_3$, the estimated cumulative loss of $O_3$ by reaction with sesquiterpenes compared to that by reaction with isoprene and monoterpenes is shown for the wet and dry seasons in Figure 2b and Figure 3b), respectively. $O_3$ reactivity ($s^{-1}$) is defined as the summed product of each terpene concentration (molec $cm^{-3}$) and its second order rate constant (molec$^{-1}$ $cm^3$ $s^{-1}$) for reaction with $O_3$. Of the approximately 20 regularly observed sesquiterpenes by SV-TAG, only two species (α-copaene and α-cedrene) have their reaction rate constants with all three major atmospheric oxidants ($O_3$, OH, and $NO_3$) measured in the laboratory (Atkinson and Arey, 2003; Shu and Atkinson, 1994). For the remaining sesquiterpenes, reaction rate constants were estimated using the Environmental Protection Agency's Estimation Program Interface Suite (U. S. EPA, 2000) or obtained from the literature where available. Reaction rate constants (estimated/measured) are listed in Table 1 for the observed sesquiterpenes. Additional reaction rate constants for other sesquiterpenes that are commonly reported but unobserved in this study are provided for comparison. The estimated contribution to $O_3$ reactivity from sesquiterpenes remains uncertain, as one to two orders of magnitude discrepancy sometimes exist between measured and estimated reaction rate constants. For example, the estimated reaction rate constant for β-caryophyllene is 44.2 x 10$^{-17}$ $cm^3$ molec$^{-1}$ $s^{-1}$ and the experimentally determined rate is 1170 x 10$^{-17}$ $cm^3$ molec$^{-1}$ $s^{-1}$, leading to calculated lifetimes via ozonolysis at 20 ppb$_v$ $O_3$ of 77 min and 3 min, respectively. Further, the estimate for monoterpene contribution to $O_3$ reactivity assumes that all monoterpenes here have same rate constant as α-pinene, as monoterpene measurements were not speciated here and it is one of the more dominant (17% by mass) and longer-lived monoterpenes emitted from the canopy (Jardine et al., 2015).

At T3, the observed sesquiterpenes are estimated to have a measurable though smaller contribution (~10-15%) to the reactive losses of $O_3$ compared to isoprene (~40%) and monoterpenes (~45%) for both seasons. The contribution of sesquiterpenes to total $O_3$ loss varies dramatically in space and time, being highest right near the sources of emission. Ozonolysis of sesquiterpenes within the canopy has been estimated to account for about 50% of sesquiterpenes' reactivity during the daytime, suggesting significant losses of these compounds before escaping the Amazon forest canopy (Jardine et al., 2011). This is reflected in our measurements of daytime minima being located far from and outside the canopy (Figure 2a and Figure 3a). This is also evident for monoterpenes, for which their concentrations and associated $O_3$ reactivity at the top of the canopy at an upwind forest site (Jardine et al., 2015) are both approximately ten times higher than those at the measurement site in this study. Further, the ratio between monoterpene and sesquiterpene concentrations typical within canopy sites upwind during the wet (7.4) and dry (2.35) seasons (Alves et al., 2016) and that observed at our measurement site for both seasons (~17) indicate that the majority of sesquiterpenes have reacted away before our measurement. This is consistent with the fact that lifetimes of all 30 sesquiterpene species detected in SV-TAG with respect to loss via reaction with $O_3$ tend to be >20 minutes, whereas more highly reactive and more commonly studied sesquiterpenes were below detection (Table 1, "Unobserved/Below Detection"). Hence, sesquiterpene contributions to $O_3$ loss in our study represents a low-end estimate of an undoubtedly important contribution of sesquiterpenes affecting ozone chemistry in the region.

### 3.2.2 Differences in chemical composition between sesquiterpene-rich essential oils and ambient samples

While it is clear that the sesquiterpenes measured at our site represent only a subset of what is emitted from regional vegetation, we further explored the compositional differences between those at our site and those in sesquiterpene-rich copaiba and andiroiba essential oils as proxies for sesquiterpene composition at the site of emission. A major difference between the ambient sesquiterpenes content and that of the essential oils, is the general absence of β-caryophyllene in ambient samples and its presence in copaiba essential oil (Figure 1 and Table S1), Andiroiba oil (Table S1), and previously analysed essential oils of Amazon origin (Table S2). The chemical reactivity of β-caryophyllene with 20 ppb$_v$ of ozone typical of polluted conditions from Manaus (Gerken et al., 2016; Trebs et al., 2012) results in a chemical lifetime of 3 minutes compared to that of α-copaene which is more than 3 hours, so any emissions are expected to be quickly depleted through reaction near the emission location. β-Caryophyllene is detected in ambient samples at our measurement site very rarely, and only when ozone is near zero ppb. This is also expected to be the case with α-humulene, which also has a chemical lifetime just under 3 minutes due to reaction





with $O_3$, though it is not as abundant as β-caryophyllene in copaiba essential oil. Some sesquiterpenes are routinely observed in ambient air that are not in the analysed copaiba essential oil (e.g. cyperene and β-gurjunene), which is expected given the rich diversity of plant species in the Amazon that should all have unique terpene contents. The difference in relative abundances of sesquiterpenes in ambient and the analysed essential oils (Table S1) also reflects many other variables:

additional vegetation with similar sesquiterpenes emission profiles, actual emission relative to tissue content, and chemical fate of the sesquiterpenes once emitted.

Based on the sesquiterpenes profile in various essential oils serving as a proxy for sesquiterpene composition upon emission, it becomes clear that the majority of potential $O_3$ reactivity from sesquiterpenes is dominated by β-caryophyllene (>80% even if only comprising approximately 20% of total sesquiterpene mass; Table S1), This demonstrates that the estimate

of $O_3$ reactivity via reactive loss with sesquiterpenes at our measurement site is a significant underestimate and not representative of near-field $O_3$ chemistry. For example, taking typical terpenoid concentrations within canopy as reported in Alves et al., 2016 and assuming monoterpene and sesquiterpene speciation within canopy to be that of Jardine et al., 2016 (for monoterpenes) and that of Copaiba Essential Oil (for sesquiterpenes), near-field or in-canopy loss of $O_3$ is dominated (> 50%) by reaction with sesquiterpenes for both seasons (Figure S1). Within the transport time to T3, the most reactive sesquiterpenes

have been reacted away, and observed contribution to $O_3$ reactivity becomes diminished by a factor of ~5 for both seasons. This analysis also highlights the importance of using speciated terpene measurements for calculating oxidative loss of radical species. For example, while the relative contributions to total terpene VOC concentrations are such that isoprene > monoterpenes > sesquiterpenes near the canopy (Figure S1) and at T3 (Figure 2a and Figure 3a), the large differences seen in terms of their relative contribution to $O_3$ reactivity at the two locations result from the terpene species prevalent at each site

(Figure S1, Figure 2b, and Figure 3b). In addition, while it is typical practice to take total sesquiterpene concentration and use $k_{O3+β\text{-caryophyllene}}$ for all sesquiterpenes (Jardine et al., 2011; Khan et al., 2017), this would result in an overestimate of sesquiterpene reactivity with $O_3$ by an order of magnitude at T3. Thus, with the majority of sesquiterpene ozonolysis occurring within/just outside the canopy, we expect to observe less sesquiterpenes at our measurement site, but that their oxidation products may be observed even when the primary sesquiterpenes are not.

**3.2.3 Observation of β-caryophyllene oxidation products**

β-Caryophyllene was infrequently observed in ambient samples despite its prevalence in copaiba essential oil, consistent with its rapid reaction with ozone during transport from the canopy to the measurement site. However, with SV-TAG we regularly observed β-caryophyllene oxidation products (specifically β-caryophyllene aldehyde and β-caryophyllonic acid) in both the gas and particle phases. These products have maxima at local night time hours (Figure 5) of at most a few ng

$m^{-3}$ (Figure S2), and exist predominantly in the gas-phase despite previous observations of these products in filter-based measurements (Chan et al., 2011; van Eijck et al., 2013; Jaoui et al., 2003; Li et al., 2011; Winterhalter et al., 2009). Based on saturation vapour pressure estimates for several β-caryophyllene oxidation products presented in Li et al., 2011 the gas-phase only observation of β-caryophyllene aldehyde (therein referred to as P-236, $C^* = 4.0 \times 10^3 \, \mu g \, m^{-3}$) is consistent with the typical organic loadings of sampled air at T3 (~1 $\mu g \, m^{-3}$ during wet season), though β-caryophyllonic acid (therein referred to

as P-252-5, $C^* = 8.7 \times 10^{-1} \, \mu g \, m^{-3}$) might be expected to have some contributions to the particle phase.

Speciation of additional oxidation products β-nocaryophyllonic acid and β-caryophyllinic acid (typically observed at sub ng/m³ levels, Figure S2) were obtained from UHPLC-qTOF-MS analysis of selected filters. These products were observed to have daily maxima during local daytime hours (Figure 5). The differing diel profiles between the four observed β-caryophyllene oxidation products likely reflects differences in chemical lifetimes of each product and multiple possible

reaction pathways of formation (i.e. β-caryophyllene initiated oxidation by $O_3$ followed by continued ozonolysis or continued OH oxidation during daytime). Further, both of these acids are estimated to have atmospheric lifetimes 2–10 days (Nozière et al., 2015), consistent with the flatter diel profile observed. In addition, because the filter-based measurements have



### 3.3 Contribution of sesquiterpene oxidation to secondary organic aerosol

### 3.3.1 Identification and quantification of common terpenoid SOA tracers

5       Here, we roughly examine the contribution of isoprene and terpene oxidation to total submicron organic aerosol (OA). Figure 6 shows a selected timeline from the wet season of the total observed OA from the AMS (averaged over the same time frame as our filter samples) and the summed contributions of positively identified molecular tracers from isoprene, monoterpene, and sesquiterpene oxidation. This analysis combines observed tracers from both SV-TAG (particle phase measurements, averaged to 12-hour time resolution to match the filter sampling times) and the UPLC filter-based

measurements of monoterpene and sesquiterpene-derived tracers, as listed in Table 2. Estimated contributions to total OA from oxidation of isoprene has a lower limit of SOA formed through the uptake of isoprene epoxydiol (IEPOX) based on positive matrix factorization (PMF) analysis of AMS data (de Sá et al., 2017). This statistical factor, known as the IEPOX-SOA factor, represents OA mass formed from isoprene under sufficiently low $NO_x$ conditions such that IEPOX forms in the gas phase (Paulot et al., 2009), followed by uptake to the particle phase to form SOA (Lin et al., 2012; Ngyuen et al., 2014;

Liu et al; 2015); it is estimated to account for approximately half of total isoprene-derived SOA in environments with low levels of $NO_x$ (Liu et al., 2015). Tracers associated with the IEPOX channel of SOA formation include 2-methyltetrols ("2-MTs") observed previously in the Amazon by Claeys et al., 2004 and C5-alkene triols (Surratt et al., 2006b, 2010), both of which are measured by SV-TAG (Isaacman-VanWertz et al., 2016) and correlate well with OA mass attributed to the AMS IEPOX-SOA factor (de Sá et al., 2017). For this reason they are presented in Figure 6 as separate from the remaining IEPOX-

SOA factor mass (i.e. IEPOX-SOA factor subtracting SV-TAG measured 2-methyltetrols and $C_5$-alkene triols). In the presence of $NO_x$, there is also production of 2-methylglyceric acid, another tracer of isoprene chemistry (Surratt et al., 2006a) measured at the molecular level by SV-TAG and also presented in Figure 6. Lower-limit estimates to total OA from oxidation of monoterpenes comprises MT filter tracers (DTAA (Iinuma et al., 2009), MBTCA (Szmigielski et al., 2007), pinic acid (Yu et al., 1999), pinonic acid (Yu et al., 1999), and terpenylic acid (Claeys et al., 2009)). Finally, lower-limit estimates to total OA

from oxidation of sesquiterpenes includes the two filter-quantified β-caryophyllene oxidation products (β-nocaryophyllonic acid and β-caryophyllinic acid). Only approximately 20% of all OA mass is composed of these identified tracer species/statistical factor, almost all of which (18%) is represented by known IEPOX products. However, the yields of known tracer compounds typically represent only a minor fraction of total terpene SOA in laboratory studies: MT acid tracers used here contribute <10% of monoterpene SOA (Kristensen et al., 2016), and the SQT acid tracers contribute <5% of sesquiterpene

SOA (van Eijck et al., 2013; Jaoui et al., 2003). Significant work therefore remains for the scientific community to resolve and identify a large fraction of terpene products to understand sources and formation processes. However, these laboratory studies provide an opportunity to estimate total contribution of monoterpene and sesquiterpene oxidation products to OA using an approximate tracer-based scaling method.

          Following the approach of (Kleindienst et al., 2007), we take the average summed concentration of β-
nocaryophyllonic acid and β-caryophyllinic acid (0.51 ng m$^{-3}$) and scale by their summed product yields (0.045) from laboratory ozonolysis of β-caryophyllene (Jaoui et al., 2003). This suggests that the contribution of β-caryophyllene to typical wet season OA concentrations of 1000 ng m$^{-3}$ is approximately 1%. However, β-caryophyllene represents only a minor fraction (median: 22%, range 6-83%) of the total sesquiterpene content of analysed essential oil/plant tissues (Table S2). This suggests that the sesquiterpene SOA contribution should be approximately 5% (range 1-18%). This is only a rough low-end estimate

because: 1) it derives from scaling two molecular tracers of β-caryophyllene oxidation; the actual SOA yield of β-caryophyllene and other sesquiterpenes oxidation is not known under conditions relevant to the Amazon, and yields likely vary as a function of air pollution and other environmental variables. Further, vapour-phase wall losses of the more semi-volatile





and lower-volatility oxidation products such as these in environmental chambers may result in non-atmospherically relevant phase partitioning and underestimated product and SOA yields (Krechmer et al., 2016; La et al., 2016; Loza et al., 2010; Matsunaga and J., 2010; McMurry and Grosjean, 1985; McVay et al., 2014; Ye et al., 2016; Yeh and Ziemann, 2014; Zhang et al., 2014, 2015). 2) This estimate derives from the available analysis of three essential oil types derived from Amazonian

tree species: copaiba essential oil (this study; Chen et al., 2009a; Soares et al., 2013), rosewood essential oil (Fidelis et al., 2012), and andiroiba essential oil (this study). While these essential oils typically comprise on order ~30 sesquiterpenes, Courtois et al., 2009 report 169 sesquiterpenes emitted by 55 species of tropical trees. Thus, the impact of additional sesquiterpenes reported to be emitted from these plants but not found in the essential oils presented here is unaccounted for. Nevertheless, this estimate demonstrates that sesquiterpene oxidation contributes measurably to SOA based on scaling from

specific identified tracers. In addition, previous analysis for the region utilizing PMF during the AMAZE-08 campaign suggests that 50% of freshly produced secondary organic material may derive from isoprene, 30% from monoterpenes, and 20% from sesquiterpenes (Chen et al., 2015). This suggests that a considerable fraction of OA from sesquiterpene oxidation remains to be accounted for through speciated measurements. Considering the chemical diversity of the sesquiterpenes observed here, it would be most ideal to have additional representative tracers and authentic standards to perform a more

accurate scaling estimate. While this remains a challenge due to the enormous range of sesquiterpenes and their oxidation products, we provide new tracers and data below.

### 3.3.2 Additional identification of terpene oxidation products by mass spectral matching

It is clear that the majority of oxidation products and SOA mass from β-caryophyllene and other sesquiterpenes (i.e. beyond β-nocaryophyllonic acid and β-caryophyllinic acid) were not identified, and thus a vast array of additional oxidation

products must be present. To explore the contributions of additional sesquiterpene oxidation products to OA, non-targeted chemical characterization was performed on selected filters from IOP1 that were collected during times when sesquiterpenes were prevalent. The compounds observed in these analyses were uniquely identified by their first and second dimension retention times, and their mass spectra. The observed compounds in ambient air samples were compared to the laboratory-generated SOA compounds from sesquiterpene oxidation, as well as a subset of known products from isoprene and

monoterpene oxidation.

Figure 7 is a GCxGC chromatogram of a night time filter sample taken Feb. 12, 2014 18:30-06:15 LT (22:30-10:15 UTC) when sesquiterpenes were relatively abundant. Approximately 460 sample peaks were separated in the chromatogram and their mass spectra and retention indices searched using NIST MS Search via GC Image (Zoex, LLC). A table of the peaks in this sample that could be attributed as BVOC oxidation products, their compound names (if positively identified), and their

assigned source category is available in Table S3. Source categories and number of peaks, n, assigned to each category in the present analysis include the following: isoprene-derived oxidation products (ISOPOX, n = 6), monoterpene-derived oxidation products (MTOX, n = 13), sesquiterpenes-derived oxidation products (SQTOX, n = 41), terpene-derived oxidation products (TERPOX, n = 10), and other organic aerosol constituents (some not easily categorized) as BVOC oxidation products (Other OA, n = 385). The TERPOX category represents compounds that are suspected to derive from monoterpene and/or

sesquiterpene oxidation based on MS similarity. That is, good MS matches include oxidation products observed in samples of laboratory monoterpene and sesquiterpene oxidation experiments (some may be overlapping products), but there was no positive delineation between MTOX or SQTOX. Categorized peaks accounted for 45% of total signal for this sample, the rest is labelled as Other OA.

The pie chart inset of Figure 7 shows the % of the sample signal (Total Ion Chromatogram, TIC, corrected for

compound transmission efficiency) associated with each source category. The peaks associated with SQTOX account for 9% of the corrected TIC, and the peaks associated with TERPOX account for another 5%, at least part of which are likely from sesquiterpene oxidation. This sample highlights the chemical complexity still to be elucidated and quantified in forthcoming





analyses of these filter samples to constrain source contributions to total SOA, while demonstrating that numerous unidentified oxidation products derived from sesquiterpenes in this region are present during local nighttime hours. This is consistent with sesquiterpenes dominating ozone reactivity during nighttime hours (Figure 2b and Figure 3b).

### 3.3.3 Identification of terpene oxidation products by chemical formulae

Five filter samples from IOP1 (wet season) during the period Feb 09 18:30 LT to Feb 11 18:30 LT were analysed according to procedures described in Section 2.3.4 Nanoelectrospray (nanoESI) Ultra-High Resolution Mass Spectrometry (UHRMS) to provide additional insight into the chemical identity (by chemical formula) of sample compounds. The presence of ions with chemical formulae consistent with oxidation products identified in the chromatography-MS methods (SV-TAG, GCxGC HR-TOFMS, and UPLC-MS) provides additional support for their prevalence throughout the wet season and is

presented in Table 2 for β-caryophyllene oxidation products and Table 3 for monoterpene oxidation products. Note that this ESI-UHRMS method is not sensitive to the isoprene oxidation products specified in Table 3, thus entries of not applicable (N/A) for observation of their chemical formulae are used. An average % of signal intensity for each UHRMS m/z is also presented for reference, but these should not be directly interpreted as the % of total OA. Rather these are the % of sample that this technique is sensitive to, as these samples have complex matrices of organic species which also affect ionization

efficiencies.

### 4 Conclusion

       We have provided speciated time-resolved measurement of 30 sesquiterpenes, four diterpenes, and have observed a broad array of sesquiterpene oxidation products in Amazonia, demonstrating that the emitted sesquiterpenes and their oxidation products in this region are both relatively abundant and highly chemically diverse. Most of the observed oxidation products

have yet to be fully chemically characterized and quantified, which will be the subject of forthcoming publications. Our observations provide a low-end estimate of the sesquiterpenes in the atmosphere closer to the forest and suggest that sesquiterpene oxidation via ozonolysis is likely the primary reactive fate of these compounds in the region. They exhibit nighttime maxima, anti-correlated with ozone, and contribute at least 10% to reactive loss of $O_3$ compared to that from reaction with isoprene and monoterpenes at our measurement site and >50% within/near the canopy. As ozone levels in this region are

directly influenced by the outflow of the Manaus plume, we would expect contributions of sesquiterpene-derived SOA to also depend highly on anthropogenic activities. In addition, since ozone enhancements at ground-level can result from downdrafts of convective storms, sesquiterpenes are part of an intricate aerosol-oxidant-cloud life cycle. Based on the observation of two β-caryophyllene oxidation products in aerosols, we estimate that sesquiterpene oxidation contributes at least 1-18% (median 5%) of observed submicron organic aerosol. As the within/near canopy reactive loss of sesquiterpenes to $O_3$ is significant, the

measurements at T3 cannot account for the most reactive sesquiterpene species, and their contributions to SOA formation in the region likely remain underestimated by the estimates reported here.

       Because several of the observed sesquiterpenes have not been studied in terms of their reaction kinetics with various oxidants (i.e. $O_3$, OH, $NO_3$) or their oxidative pathways leading to SOA, it is still challenging to fully constrain their role in the atmospheric chemistry of the region. Part of this challenge is stymied by the lack of available authentic standards, so

further work in separation of complex mixtures (e.g. essential oils) to isolate pure sesquiterpenes and chemical synthesis of oxidation products would prove beneficial for furthering our knowledge of sesquiterpene chemistry. Future field work should focus on performing speciated sesquiterpene measurements within the canopy and connecting the chemical fate and transport from emission to regional atmospheric chemistry. Further, laboratory oxidation experiments of newly-observed sesquiterpenes or relevant mixtures could be used to improve estimates of reactive oxidant loss and contributions to SOA.



**Data availability**

The data sets used in this publication are available at the ARM Climate Research Facility database for the GoAmazon2014/5 campaign (https://www.arm.gov/research/campaigns/amf2014goamazon).

**Competing Interests**

The authors declare that they have no conflict of interest

**Author contribution**

AG, LM, RdS, AM, PA, JJ, and SM designed, coordinated, and supervised the GoAmazon field campaign and LY, GI, RW, NK, SV, SdS, LA, BB, WH, PC, DD, YL, KM, JV, MO, and KL carried out the measurements and model simulations. LY, GI, MM, VR, RW, SdS, BB, and YL performed data analysis. MB, MG, IK, and MK performed additional analysis of

collected filter samples. MA, AB, RT, and FG synthesized and provided chemical standards. JO and ML provided supplementary filter samples from laboratory oxidation experiments to aid data interpretation. LY prepared the manuscript with contributions from all co-authors.

**Acknowledgements**

The UC Berkeley team was supported for the GoAmazon2014/15 field campaign by NSF ACP Grant #1332998, and for further

analysis of the dataset by DOE ASR Grant #DE-SC0014040. The Northwestern University team was supported by the National Science Foundation (NSF) under grant no. CHE-1607640. The instrument as deployed was developed through support from U.S. Department of Energy (DOE) SBIR grant DE-SC0004698. We gratefully acknowledge support from the Central Office of the Large Scale Biosphere Atmosphere Experiment in Amazonia (LBA), the Instituto Nacional de Pesquisas da Amazonia (INPA), and the Universidade do Estado do Amazonia (UEA) and the local Fundation (FAPEAM). The work was conducted

under 001030/2012-4 of the Brazilian National Council for Scientific and Technological Development (CNPq). We acknowledge logistical support from the Atmospheric Radiation Measurement (ARM) Climate Research Facility, a U.S. Department of Energy Office of Science user facility sponsored by the Office of Biological and Environmental Research. ARM-collected data including ozone and meteorology were obtained from MAOS. L. D. Y. acknowledges support from a University of California Berkeley Chancellor's Postdoctoral Fellowship. G. I. V. W. acknowledges support from a NSF

Graduate Research Fellowship (#DGE 1106400). B. B. P. acknowledges support from a U.S. EPA STAR Graduate Fellowship (FP-91761701-0). The University of Colorado group acknowledges support from DOE (BER/ASR) DE-SC0016559 and EPA-STAR 83587701-0. EPA has not reviewed this manuscript, and thus no endorsement should be inferred. A. G. B. and M. A. U. gratefully acknowledge support from NSF Graduate Research Fellowships. M. A. U. also acknowledges a NSF GROW award, National Aeronautics and Space Administration Earth and Space (NASA ESS) Fellowship and a P. E. O. Scholar

Award. F. M. G. gratefully acknowledges support from the Alexander von Humboldt Foundation.

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



**Tables**

Table 1. Sesquiterpene and terpenoids observed in the gas phase during GoAmazon2014/5 with proposed identification and their alkane-based retention index, Chemical Abstract Service (CAS) #, mean concentration during wet/dry season, reaction rate constant with $O_3$ ($k_{O3}$), and estimated chemical lifetime in the presence of 20 ppbᵥ $O_3$. $k_{O3}$ estimated using EPA EPI Suite 4.1 AOPWIN where
5 literature data unavailable. Concentrations of each species estimated by using an average instrument response factor for several sesquiterpene standards unless otherwise noted that an authentic standard was used. Other commonly studied sesquiterpenes typically below detection or unobserved are also included for comparison of reactive timescales.

| # | Compound | Retention Index | CAS # | Wet Season Mean Concentration | | Dry Season Mean Concentration | | $k_{O3}$ x $10^{17}$ (cm$^3$ molec$^{-1}$ s$^{-1}$) | $\tau$ for [$O_3$] = 20 ppbᵥ (min) |
|---|---|---|---|---|---|---|---|---|---|
| | | | | (ng m$^{-3}$) | (ppqᵥ) | (ng m$^{-3}$) | (ppqᵥ) | | |
| **Sesquiterpenes** | | | | | | | | | |
| 1 | α-cubebene | 1355 | 17699-14-8 | 2.25 | 250 | 1.90 | 211 | 43.0 | 78.9 |
| 2 | unidentified | 1380 | | 2.86 | 318 | 1.12 | 125 | | |
| 3 | cyclosativene | 1383 | 22469-52-9 | 2.86 | 317 | 1.16 | 129 | 7.4 | 458.7 |
| 4 | α-copaene | 1387 | 3856-25-5 | 3.75[1] | 417 | 3.92[1] | 435 | 16[2] | 212.0 |
| 5 | β-elemene | 1397 | 515-13-9 | 2.08 | 231 | 1.46 | 163 | 2.6 | 1317.2 |
| 6 | cyperene | 1420 | 2387-78-2 | 1.61 | 179 | 0.81 | 90 | NE[3] | |
| 7 | α-cedrene | 1435 | 469-61-4 | 0.25[1] | 28 | 0.45[1] | 49 | 2.8[2] | 78.9 |
| 8 | unidentified | 1443 | | 4.76 | 529 | 6.04 | 670 | | |
| 9 | β-gurjunene | 1447 | 17334-55-3 | 0.88 | 97 | 0.88 | 98 | 7.4 | 458.7 |
| 10 | unidentified | 1453 | 26620-71-3 | 0.32 | 35 | 0.01 | 2 | | |
| 11 | unidentified | 1462 | | 1.61 | 179 | 1.38 | 153 | | |
| 12 | α-patchoulene | 1471 | 560-32-7 | 0.27 | 30 | 1.01 | 113 | 7.4 | 458.7 |
| 13 | unidentified; SQT202 | 1474 | | 0.29 | 33 | 0.43 | 49 | | |
| 14 | (-) alloaromadendrene[1] | 1476 | 025246-27-9 | 1.93[1] | 214 | 1.72[1] | 191 | 1.2 | 2826.4 |
| 15 | γ-muurolene | 1486 | 30021-74-0 | 0.18 | 20 | 0.11 | 12 | 44.2 | 76.7 |
| 16 | unidentified; SQT202 | 1486 | | 2.36 | 265 | 4.89 | 548 | | |
| 17 | α-amorphene | 1490 | 20085-19-2 | 1.23 | 136 | 1.03 | 114 | 86.0 | 39.4 |
| 18 | β-selinene | 1505 | 17066-67-0 | 0.45 | 50 | 2.38 | 264 | 2.4 | 1413.2 |
| 19 | α-muurolene | 1509 | 31983-22-9 | 1.50 | 166 | 1.45 | 161 | 86.0 | 39.4 |
| 20 | unidentified | 1512 | | 2.14 | 237 | 1.50 | 167 | | |
| 21 | β-bisabolene | 1513 | 495-61-4 | 1.24 | 138 | 0.32 | 36 | 87.20 | 38.9 |
| 22 | cuparene | 1524 | 16982-00-6 | 1.13 | 127 | 0.62 | 69 | | |
| 23 | γ-cadinene | 1526 | 39029-41-9 | 1.75 | 194 | 1.18 | 131 | 44.2 | 76.7 |
| 24 | δ-cadinene | 1529 | 483-76-1 | 0.95 | 105 | 0.81 | 90 | 163.0 | 20.8 |
| 25 | cis-calamenene; | 1534 | 72937-55-4 | 0.81 | 91 | 0.32 | 35 | NE[3] | |
| 26 | selinene <7-epi-α> | 1537 | 6813-21-4 | 0.14 | 15 | 0.22 | 25 | 163.0 | 20.8 |
| 27 | γ-cuprenene | 1545 | 4895-23-2 | 0.32 | 35 | 0.14 | 15 | 50.4 | 67.3 |
| 28 | α-cadinene | 1549 | 24406-05-1 | 0.17 | 18 | 0.06 | 7 | 86.0 | 39.4 |
| 29 | unidentified | 1553 | | 0.01 | 1 | 0.13 | 14 | | |
| 30 | Selina-3,7(11)-diene | 1558 | 6813-21-4 | 0.07 | 8 | 0.10 | 11 | 163.0 | 20.8 |
| **Below Detection/Unobserved** | | | | | | | | | |
| | β-caryophyllene | 1427 | 87-44-5 | N/A | N/A | N/A | N/A | 1160.0 | 2.9 |
| | trans-α-bergamotene | 1442 | 13474-59-4 | N/A | N/A | N/A | N/A | 86.0 | 39.4 |
| | aromadendrene | 1450 | 489-39-4 | N/A | N/A | N/A | N/A | 1.2 | 2826.4 |
| | α-humulene | 1471 | 6753-98-6 | N/A | N/A | N/A | N/A | 1170.0 | 2.9 |
| | β-farnesene | | 18794-84-8 | N/A | N/A | N/A | N/A | 40.1[4] | 84.6 |
| | α-farnesene | 1509 | 502-61-4 | N/A | N/A | N/A | N/A | 104.0 | 32.6 |
| | valencene | 1523 | 4630-07-3 | N/A | N/A | N/A | N/A | 8.6 | 394.7 |
| **Diterpenes** | | | | | | | | | |
| 31 | rimuene | 1958 | 1686-67-5 | 0.21 | 18 | 0.22 | 19 | 7.6 | 448.1 |
| 32 | pimaradiene | 1977 | 1686-61-9 | 0.11 | 10 | 0.14 | 12 | 7.6 | 448.1 |
| 33 | Sandaracopimaradiene | 1995 | 1686-56-2 | 0.38 | 33 | 0.15 | 13 | 7.6 | 448.1 |
| 34 | kaurene | 2085 | 34424-57-2 | 0.97 | 86 | 0.69 | 60 | 1.1 | 2981.7 |
| | **Total Sesquiterpenes + Diterpenes:** | | | **41.8** | **4611** | **38.8** | **4285** | **N/A** | **N/A** |

[1] Authentic standard used for quantification
[2] (Shu and Atkinson, 1994)
[3] NE=No estimate in EPISuite
[4] (Kourtchev et al., 2009)





**Table 2: Oxidation products from beta-caryophyllene observed in Central Amazonia in gas and/or particle phases by listed analysis method. These products are attributed to oxidation from sesquiterpenes (SQTOX) source category. Synthesized standards for tracers 1–6 were used to confirm identification and for quantification.**

| # | Tracer Name | Structure | Source Category | Analysis Methods | Chemical Formula | Molecular Weight (g/mol) | UHRMS Measured m/z [M-1]⁻ | UHRMS Average % of Total Signal Intensity |
|---|---|---|---|---|---|---|---|---|
| 1 | β-caryophyllene aldehyde | | SQTOX | SV-TAG; Filter GCxGC HR-TOF-MS; Filter ESI-UHRMS | $C_{15}H_{24}O_2$ | 236.35 | 235.17023 | 1.2 |
| 2 | β-caryophyllonic acid | | SQTOX | SV-TAG; Filter GCxGC HR-TOF-MS; Filter ESI-UHRMS | $C_{15}H_{24}O_3$ | 252.35 | 251.1652 | 0.74 |
| 3 | β-nocaryophyllone aldehyde | | SQTOX | Filter GCxGC HR-TOF-MS; Filter ESI-UHRMS | $C_{14}H_{22}O_3$ | 238.32 | 237.14943 | 0.37 |
| 4 | β-nocaryophyllonic acid | | SQTOX | Filter UPLC-MS; Filter ESI-UHRMS | $C_{14}H_{22}O_4$ | 254.32 | 253.14448 | 0.81 |
| 5 | β-caryophyllinic acid | | SQTOX | Filter UPLC-MS; Filter ESI-UHRMS | $C_{14}H_{22}O_4$ | 254.32 | 253.14448 | 0.81 |
| 6 | β-nocaryophyllinic acid | | SQTOX | Filter ESI-UHRMS | $C_{13}H_{20}O_5$ | 256.29 | 255.12378 | 0.69 |
| 7 | DCCA or 3,3-dimethyl-2-(3-oxobutyl)-cyclobutanecarboxylic acid | | SQTOX | Filter ESI-UHRMS | $C_{11}H_{18}O_3$ | 198.26 | 197.11828 | 1.25 |
| 8 | 2-(2-carboxyethyl)-3,3-dimethylcyclo-butanecarboxylic acid | | SQTOX | Filter ESI-UHRMS | $C_{10}H_{16}O_4$ | 200.23 | 199.09755 | 1.36 |





Table 3: Oxidation products from isoprene (ISOPOX) and monoterpenes (MTOX) in Central Amazonia in gas and/or particle phases. Only particle-phase measurement presented in this study. Products marked not applicable (N/A) are not detected in the UHRMS analysis method.

| # | Tracer Name | Source Category | Analysis Methods | Chemical Formula | Molecular Weight (g/mol) | UHRMS Measured m/z [M-1]⁻ | UHRMS Average % of Total Signal |
|---|---|---|---|---|---|---|---|
| 1 | 2-MT 1 or 2-methylerythritol | ISOPOX | SV-TAG; Filter GCxGC HR-TOF-MS | $C_5H_{12}O_4$ | 136.15 | N/A | N/A |
| 2 | 2-MT 2 or 2-methylthreitol | ISOPOX | SV-TAG; Filter GCxGC HR-TOF-MS | $C_5H_{12}O_4$ | 136.15 | N/A | N/A |
| 3 | C5 alktriol 1 or 3-methyl-2,3,4-trihydroxy-1-butene | ISOPOX | SV-TAG; Filter GCxGC HR-TOF-MS | $C_5H_{10}O_3$ | 118.13 | N/A | N/A |
| 4 | C5 alktriol 2 or cis-2-methyl-1,3,4-trihydroxy-1-butene | ISOPOX | SV-TAG; Filter GCxGC HR-TOF-MS | $C_5H_{10}O_3$ | 118.13 | N/A | N/A |
| 5 | 2-MGA or 2-methylglyceric acid | ISOPOX | SV-TAG; Filter GCxGC HR-TOF-MS | $C_4H_8O_4$ | 120.10 | N/A | N/A |
| 6 | DTAA or diaterpenylic acid acetate | MTOX | Filter UPLC-MS; Filter ESI-UHRMS | $C_{10}H_{16}O_6$ | 232.23 | 231.08728 | 0.57 |
| 7 | MBTCA or 3-methyl-1,2,3-butanetricarboxy | MTOX | Filter GCxGC HR-TOF-MS; Filter UPLC-MS; Filter ESI-UHRMS | $C_8H_{12}O_6$ | 204.18 | 203.05606 | 0.87 |
| 8 | Pinic acid | MTOX | Filter GCxGC HR-TOF-MS; Filter UPLC-MS; Filter ESI-UHRMS | $C_9H_{14}O_4$ | 186.21 | 185.08186 | 0.96 |
| 9 | Pinonic acid | MTOX | Filter UPLC-MS; Filter ESI-UHRMS | $C_{10}H_{16}O_3$ | 184.23 | 183.10259 | 2.73 |
| 10 | Terpenylic acid | MTOX | Filter UPLC-MS; Filter ESI-UHRMS | $C_8H_{12}O_4$ | 172.18 | 171.06618 | 0.78 |
| 11 | cis-norpinic acid | MTOX | Filter GCxGC HR-TOF-MS; Filter ESI-UHRMS | $C_9H_{16}O_3$ | 172.22 | 171.10252 | 2.97 |





**Figures**

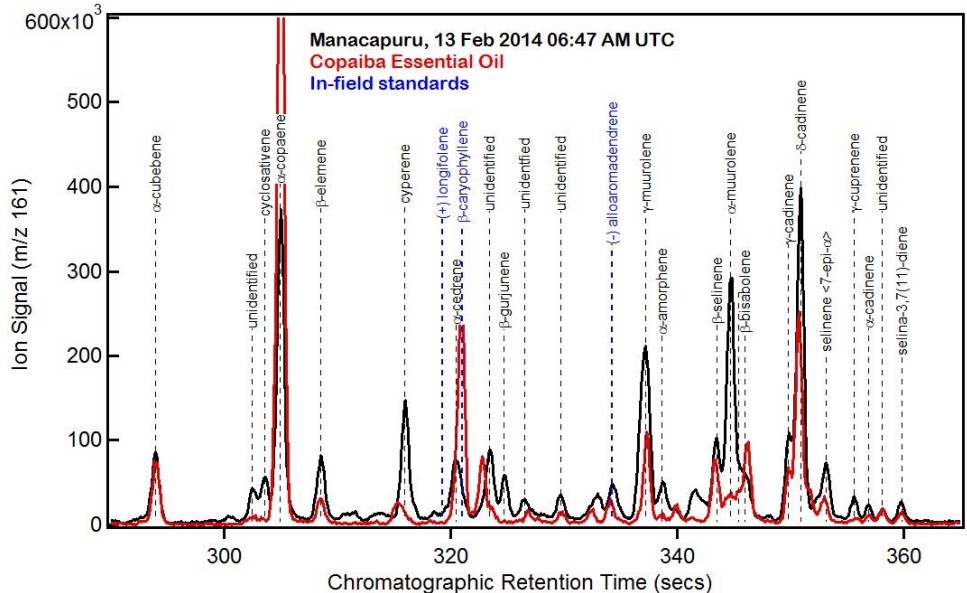

**Figure 1: Selected ion chromatograms at sesquiterpene characteristic ion *m/z* 161 of ambient air (black) and copaiba essential oil (red). Sesquiterpenes in ambient air were measured in the gas phase. Copaiba essential oil analysed from direct liquid injection on**
5 **SV-TAG collection cells. Retention times for standards analysed in the field indicated in blue.**

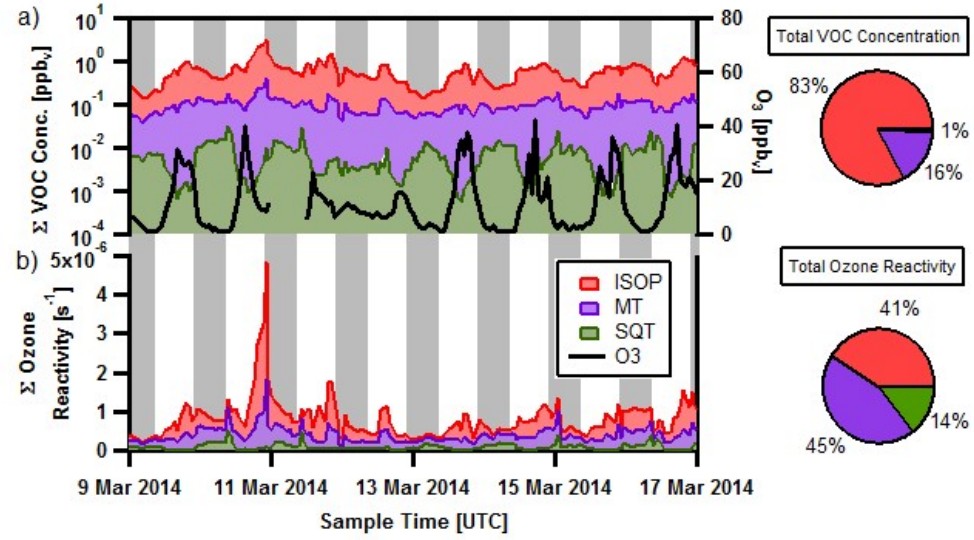

**Figure 2: Wet season selected timeline of summed gas-phase VOC concentrations, isoprene (ISOP, red), monoterpenes (MT, purple), and estimated sesquiterpenes (SQT, green) and ozone concentration (O₃, black) in panel a). Average % contributions during wet**
10 **season for each group to total VOC concentration during also shown in top pie chart. Panel b) depicts summed contribution to ozone reactivity from isoprene, monoterpenes, and sesquiterpenes. Average % contributions during wet season for each VOC to total ozone reactivity shown in bottom pie chart. Local nighttime hours indicated in grey. Local time is -4 h relative to UTC.**





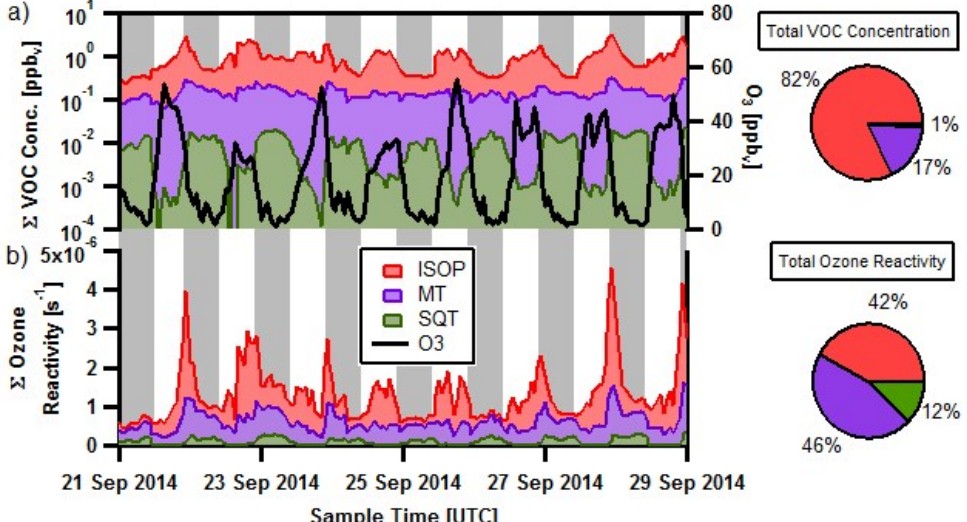

**Figure 3: Dry season selected timeline of summed gas-phase VOC concentrations, isoprene (ISOP, red), monoterpenes (MT, purple), and estimated sesquiterpenes (SQT, green) and ozone concentration (O₃, black) in panel a). Average % contributions during dry season for each group to total VOC concentration during also shown in top pie chart. Panel b) depicts summed contribution to ozone reactivity from isoprene, monoterpenes, and sesquiterpenes. Average % contributions during dry season for each VOC to total ozone reactivity shown in bottom pie chart. Local nighttime hours indicated in grey. Local time is -4 h relative to UTC.**



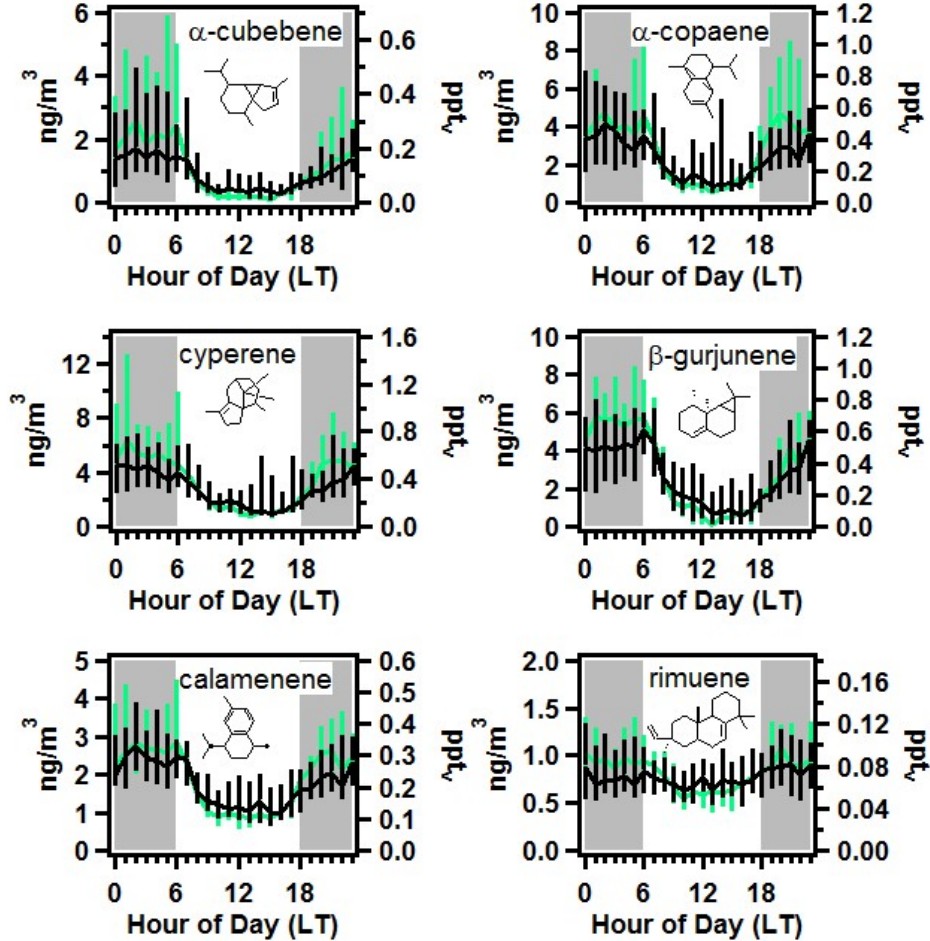

**Figure 4: Average diel profiles of selected sesquiterpenes and one diterpene (rimuene) for wet season (light green) and dry season (black) in gas phase. Solid line is drawn through median values, and bars indicate range from 25- to 75- percentiles. Local night**
5 **time hours are indicated in grey shading.**



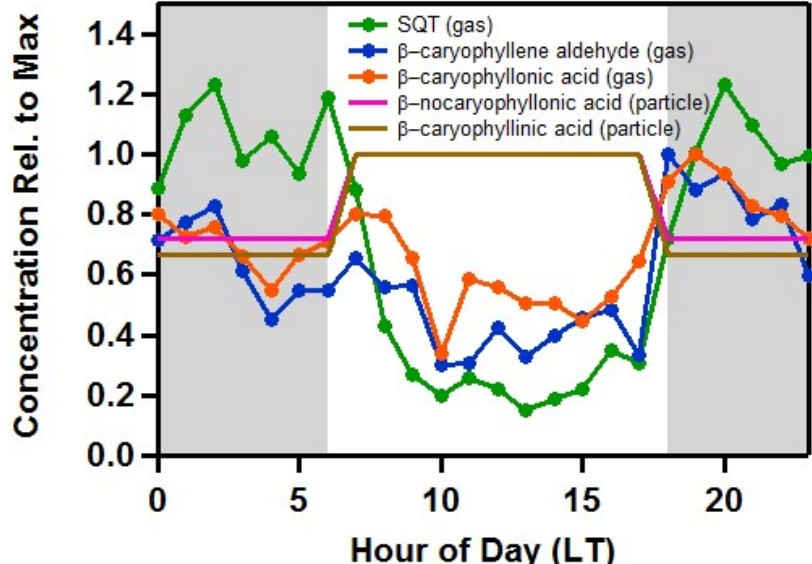

**Figure 5: Diel profiles during wet season of total sesquiterpenes (SQT) and four β-caryophyllene oxidation products in gas and particle phases. Local night time hours indicated in grey.**

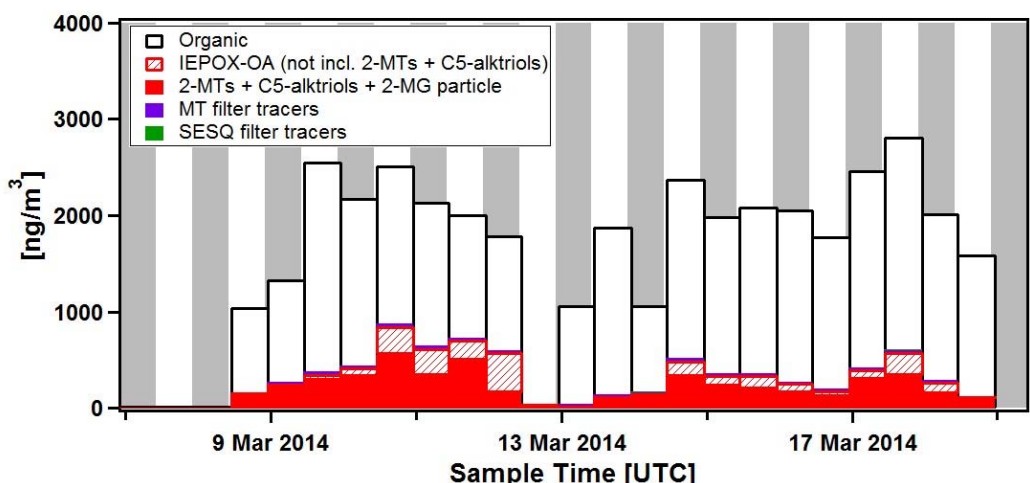

**Figure 6: Estimated contributions to total organic aerosol from particle-phase tracers/statistical factors attributed to oxidation of isoprene (red) (IEPOX-SOA, 2-MTs + C5alktriols + 2-MG), monoterpene (purple) (MT filter tracers), and sesquiterpene (green) (SESQ filter tracers). 2-MTs = 2-methyl tetrols, 2-MG = 2-methylglyceric acid, and C5alktriols is C5-alkene triols. Local night time**
10  **hours indicated in grey.**



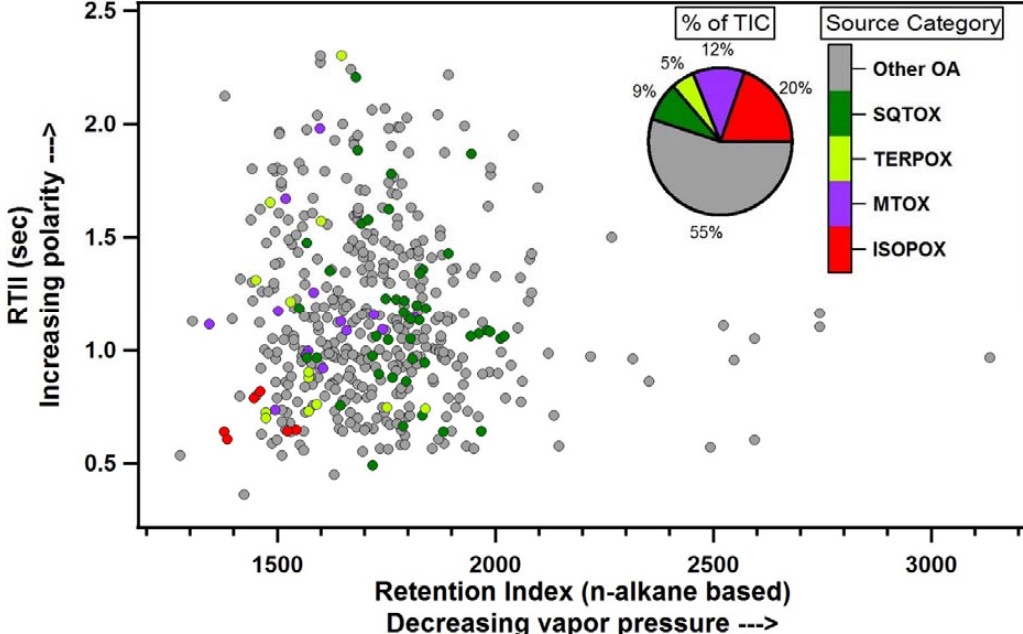

**Figure 7: GCxGC Chromatogram of nighttime filter sample during wet season. Peaks are assigned source categories of isoprene oxidation products (ISOPOX) in red, monoterpene oxidation products (MTOX) in purple, terpene oxidation products (TERPOX) in yellow, and sesquiterpene oxidation products (SQTOX) in green. Other unidentified OA in gray. % of total ion chromatogram made up by each source category shown in pie chart.**