# Peer review of "Observations of sesquiterpenes and their oxidation products in central Amazonia during the wet and dry seasons"

_Atmospheric Chemistry and Physics, 2018_

## Referee Comment (RC1) · Anonymous Referee #1 · 17 Apr 2018

This manuscript describes observations of sesquiterpenes and their oxidation products in gas- and particle-phase samples during a field campaign in the Amazon. This is a topic of interest to the readership of Atmospheric Chemistry and Physics due to recent focus on the contribution of biogenic volatile organic compounds (BVOCs) to ozone and secondary organic aerosol formation, 'missing' BVOC reactivity in forest environments, and the processes that occur when BVOCs are emitted into air that has been influenced by anthropogenic emissions.

The paper is appropriately cited, and the abstract clearly conveys the content of the manuscript. All figures and tables are necessary and appropriate.

[Figure]

The manuscript is clearly written. Only minor editing is required; minor comments of this nature are included below the Technical Comments.

There are relatively minor technical changes/clarifications that are needed before this manuscript can be published. These also are listed below.

Technical Comments

1. In section 2.2, what was done to ensure that the denuder efficiency was 100%? Or was this efficiency calculated in previous publications? If so, please provide the number and appropriate citation - and if appropriate, discuss how this efficiency was taken into account during quantification.

2. In section 2.2.1, please define the match statistic. It is clear that a maximum value is 999, but it is not clear how the values are obtained.

3. In some cases, abbreviations/acronyms/chemical symbols are defined prior to use. In others, they are not. In some cases, they are not used consistently. Please make this consistent.

4. In section 2.2.3, Tables S4 and S5 are cited – but Tables S1-S3 have not been called out yet. This is simply a matter of reordering the tables in the SI.

5. On page 7, line 14. Can the authors explain why dry season filter samples are not included here? It seems odd that wet and dry season SVTAG output is included, but only wet season filter samples are. Do these samples not exist? It would strengthen the paper greatly if both season's filter results were included, allowing comparison of seasons and discussion of 'representativeness.'

6. On page 10, line 8, the statement 'or obtained from the literature where available' seems to contradict the previous statement about lack of availability of rate constant data. I could simply be misunderstanding, but please clarify?

7. On page 10, line 17, where transport to the site is discussed as the reason for lack of

observation of more quickly reacting sesquiterpenes, please discuss what the typical transport time to the site from the canopy is.

8. On Figure 5, what is the maximum to which the concentrations are normalized? The aldehyde product? The acid product? Both have values that are ∼1 around sundown.

9. In the SI, is Figure S3 called out/cited anywhere?

Editorial Corrections/Recommendations (other very minor typos should be found when proofs are reviewed)

1. In several places throughout the manuscript, the authors need to format citations that appear in the main text (as opposed to in the parentheses). For example, page 3, line 20, 'Khan et al., 2017' should be 'Khan et al. (2017)'.

2. On page 8, lines 34+, numbers in chemical formulae should be subscripted.

3. I would recommend replacing '%' with 'percentage' in several places where it appears as part of the text – for example, page 13, line 39. This is simply preference.

4. Caption to Table S3 in SI. There appears to be an extra 'in'

Caption to Figure S1 in SI. There appears to be an 'in' at the end that is not necessary.

---

## Referee Comment (RC2) · Anonymous Referee #2 · 7 May 2018

The study by Lindsay D. Yee et al. deals with sesquiterpene observations in Brazil as part of the Green Ocean Amazon (GoAmazon2014/5) campaign. The authors present measurements from a site that is not located inside the pristine Amazon rainforest and discuss the observed oxidation products both in gas and particle phase, during the dry and wet season. Their observations include a vast array of sesquiterpenes (and 4 diterpenes) that are reported to be in the ppq range which is much lower than any other measurements that have been conducted inside the forested area. Nonetheless, the measurement location has provided the opportunity to investigate oxidation products and calculate a low end O3 reactivity. It is important that throughout the text the authors openly discuss the limitations of their dataset. That said, the study is not as novel and

conclusive but I still believe that it is an important addition to the limited literature of sesquiterpenes. Overall, the manuscript is very well written and fits within the scope of the special issue submitted. I would therefore suggest publication to ACP after addressing the following issues.

General comments:

1. The calibration procedure has to be described and presented in greater detail. Please include further details on how the calibrations were performed, how often and with what technical characteristics (e.g. detection limits, precision, accuracy).

2. There has to be a section over which the observations are thoroughly presented. The results section starts with a subchapter named "Chemical characterization of observed sesquiterpenes" but there is mainly technical descriptions and not presentation of the observations. In addition, only selected time frames are presented in both manuscript and supplement. I would encourage the authors to include a complete timeline of their measurements and certainly move the largest part of this section (3.1) in the methods.

3. While the uncertainties on the reaction rate constants are discussed for sesquiterpenes, the same rate constant as a-pinene has been applied for monoterpenes. Yanez-Serrano et al. (2018) demonstrated a similar chemodiversity of monoterpenes for both wet and dry season inside the Amazon rainforest. Therefore, the uncertainties of monoterpene reactivity (and hence the relative contribution to isoprene and sesquiterpenes) can be minimized. I recommend re-calculation of the $O_3$ reactivity based on the monoterpene speciation from the literature with the respective reaction rates and relative abundance.

General technical comments:

1. Please ensure that the supplementary material is appropriately cited in the main text.

2. Please ensure that your references conform to the ACP style.

Specific comments:

P3L25-28. No need to repeat the measurement challenges as they were already mentioned above.

P4L5. You may keep the definition of IOP but it would be better if you refer to your periods as wet and dry season thereafter.

P4L17. There is no need for this last sentence.

P8L31. Please rename as "Results and discussion". As mentioned above, I would recommend to include a section over which the observations are described.

P8L36. This class of compounds is referred as sesquiterpenoids in Chan et al. (2016).

P9L24. Please site the "previous literature". P9L34-36. Did you observe such case? Is there a possibility of presenting a case study?

P10L10-14 and L30. It would be interesting if an upper end of sesquiterpene estimated O3 reactivity is presented as well.

P10L14-16. Please see my general comment.

P11L21-22. Nonetheless, this is your practice for monoterpenes. Please discuss a quantitative

Figure 6. The filter tracers are not visible. Maybe the use of log scale would help?

References:

Yáñez-Serrano, A. M., Nölscher, A. C., Bourtsoukidis, E., Gomes Alves, E., Ganzeveld, L., Bonn, B., Wolff, S., Sa, M., Yamasoe, M., Williams, J., Andreae, M. O., and Kesselmeier, J.: Monoterpene chemical speciation in a tropical rainforest:variation with season, height, and time of dayat the Amazon Tall Tower Observatory (ATTO), Atmos. Chem. Phys., 18, 3403-3418, https://doi.org/10.5194/acp-18-3403-2018, 2018.

Chan, A. W. H., Kreisberg, N. M., Hohaus, T., Campuzano-Jost, P., Zhao, Y., Day, D. A., Kaser, L., Karl, T., Hansel, A., Teng, A. P., Ruehl, C. R., Sueper, D. T., Jayne, J. T., Worsnop, D. R., Jimenez, J. L., Hering, S. V., and Goldstein, A. H.: Speciated measurements of semivolatile and intermediate volatility organic compounds (S/IVOCs) in a pine forest during BEACHON-RoMBAS 2011, Atmos. Chem. Phys., 16, 1187-1205, https://doi.org/10.5194/acp-16-1187-2016, 2016.

---

## Author Comment (AC1) · 27 Jun 2018

**Response to Reviewers for "Observations of sesquiterpenes and their oxidation products in central Amazonia during the wet and dry seasons"**

**The authors thank both referees for their helpful comments towards improving this manuscript. All referee comments are addressed below. Author comments are formatted in blue text. Page numbers and line numbers are according to the ACPD published manuscript.**

**Response to Anonymous Referee #1 Comments:**

Technical Comments

1. In section 2.2, what was done to ensure that the denuder efficiency was 100%? Or was this efficiency calculated in previous publications? If so, please provide the number and appropriate citation - and if appropriate, discuss how this efficiency was taken into account during quantification.

   We have added after the discussion pg. 4, line 36:
   As described in Isaacmann-VanWertz et al., (2016), regular checks of denuder efficiency were done by inserting a filter upstream of the denuder to remove particles, and sampling the normal volume of air through this "blank" system so the measured signal would indicate any breakthrough. Any remaining mass signal was subtracted from the sample mass signal as part of data correction before quantification. Previous laboratory testing of the denuder efficiency was also performed by sending gas standards (e.g. the sesquiterpene longifolene) through the denuder and measuring the sesquiterpene signal upstream and downstream using proton-transfer-reaction-mass-spectrometry. This led to a calculated penetration value on average of <5% for a single denuder and a predicted penetration of <0.5% for the two denuders used in series on SVTAG.

2. In section 2.2.1, please define the match statistic. It is clear that a maximum value is 999, but it is not clear how the values are obtained.

   The "match statistic" is the same as the "match factor" calculated within the NIST/EPA/NIH Mass Spectral Library program in accordance to the methods described in (Stein, 1994). To be more precise and consistent, we have changed the term to "match factor" throughout the manuscript and an additional sentence has been added in section 2.2.1, "A match factor is calculated from a comparison function outlined in Stein et al., 1994 as a measure of the overall probability that an obtained spectral match is correct. Spectral matches are considered …"

3. In some cases, abbreviations/acronyms/chemical symbols are defined prior to use. In others, they are not. In some cases, they are not used consistently. Please make this consistent.

   This has now been addressed. Thank you for noticing this.

4. In section 2.2.3, Tables S4 and S5 are cited – but Tables S1-S3 have not been called out yet. This is simply a matter of reordering the tables in the SI.

   Thank you for catching this. The tables have been properly reordered in the SI to match the order of reference in the main text.

5. On page 7, line 14. Can the authors explain why dry season filter samples are not included here? It seems odd that wet and dry season SVTAG output is included, but only wet season filter samples are. Do these samples not exist? It would strengthen the paper greatly if both season's filter results were included, allowing comparison of seasons and discussion of 'representativeness.'

   As only selected wet and dry season filter samples have been analyzed thus far (complete set will be analyzed for an upcoming publication), we merely meant to show a representative sample that would be rich in a variety of sesquiterpene oxidation products. A similarly targeted filter sample from the dry season was also analyzed and similar to the sample presented for the wet season. We do understand the question raised, so we have adjusted the text further to explain this accordingly:

   Filter samples collected during IOP2 (dry season) are not presented in this analysis as the wet season filters were more ideal for targeted isolation and detection of sesquiterpene oxidation products. Similarly targeted samples from the dry season had similar chemical composition in terms of terpene oxidation as that presented in Section 3.3 for the wet season so this presentation is not repeated, though there are certainly contributions from additional OA sources (e.g. biomass burning compounds are more prominent in dry than wet season) as well. A more complete analysis of all samples from both seasons will be presented in separate forthcoming publication. The goal in the current analysis is to simply demonstrate the number and chemical complexity of the observed sesquiterpene-derived compounds and the potential for their significance in contributing to overall OA mass.

6. On page 10, line 8, the statement 'or obtained from the literature where available' seems to contradict the previous statement about lack of availability of rate constant data. I could simply be misunderstanding, but please clarify?

   The authors acknowledge that the text is confusing here and have deleted the phrase, "or obtained from the literature where available."

7. On page 10, line 17, where transport to the site is discussed as the reason for lack of observation of more quickly reacting sesquiterpenes, please discuss what the typical transport time to the site from the canopy is.

   We have included the following sentence, "Based on average wind speed (2 m s-1), transport time from the nearest surrounding trees (1 km) to the measurement site is on the order of at

least 8 minutes, longer than the chemical lifetime of some of the more highly reactive sesquiterpenes."

8.  On Figure 5, what is the maximum to which the concentrations are normalized? The aldehyde product? The acid product? Both have values that are 1 around sundown.

    Each of these series is normalized to its own average maximum concentration observed.  This makes every series have 1 as the relative maximum concentration as plotted in Figure 5.  To be more clear, we have added text in the caption of Figure 5 to describe this, "For each series, data are normalized by the maximum observed concentration within the series and shown as concentration relative to max."

9.  In the SI, is Figure S3 called out/cited anywhere?

    We have added reference to this figure at the end of Section 2.2 describing deployment of SV-TAG to explain how continuous time series of total sesquiterpenes were generated.  As each sesquiterpene typically made up a small fraction of total ion signal in derivatized runs, we utilized the less frequent (~ every 13 hrs) runs without derivatization to speciate all sesquiterpenes and obtain total sesquiterpene concentration.

    "To generate continuous time series of total sesquiterpenes concentration as presented in section 3.2.1, we assumed that the longer-lived and regularly detected α-copaene comprised 6% of total sesquiterpenes concentration at all times, since this was the average % composition during runs without derivatization (Figure S3)."

Editorial Corrections/Recommendations (other very minor typos should be found when proofs are reviewed)

1.  In several places throughout the manuscript, the authors need to format citations that appear in the main text (as opposed to in the parentheses). For example, page 3, line 20, 'Khan et al., 2017' should be 'Khan et al. (2017)'.

    Thank you for catching this, we have adjusted all references accordingly.

2.  On page 8, lines 34+, numbers in chemical formulae should be subscripted.

    These changes have been addressed.

3.  I would recommend replacing '%' with 'percentage' in several places where it appears as part of the text – for example, page 13, line 39. This is simply preference.

    These changes have been addressed.

4.  Caption to Table S3 in SI. There appears to be an extra 'in' Caption to Figure S1 in SI. There appears to be an 'in' at the end that is not necessary.

The unnecessary "in" has been deleted.  Thank you for catching this.

**Response to Anonymous Referee #2 Comments:**

General comments:

1.  The calibration procedure has to be described and presented in greater detail. Please include further details on how the calibrations were performed, how often and with what technical characteristics (e.g. detection limits, precision, accuracy).

    We have adjusted Section 2.2.2 Compound quantification as follows:
    a)  We have started the section now with, "In-field calibrations on SV-TAG were performed using an auto liquid injection system (Isaacman et al., 2011) to deliver customized standard solutions. A calibration point was obtained every 6-7 hrs, rendering a complete six-point calibration curve within 48 hrs."
    b)  We have added pg. 5, line 38 after "…relative response factors to β-caryophyllene," the following sentence, "A range of instrument responses to sesquiterpene standards was observed. For example, on-column lower detection limits were 0.14, 0.01, 0.05, and 0.08 ng with precision of 14%, 21%, 9.5%, and 13% and accuracy of 12%, 7.4%, 25%, and 17%, for β-caryophyllene, longifolene, alloaromadendrene, and α-copaene, respectively."
    c)  We have added after the sentence pg. 6, line 2, "Calculated on-column lower detection limit is 0.07 ng with typical precision on the order of 14% and accuracy errors within 30%. Additional details of error analysis for SV-TAG data are detailed in Isaacman et al., (2014)."

2.  There has to be a section over which the observations are thoroughly presented. The results section starts with a subchapter named "Chemical characterization of observed sesquiterpenes" but there is mainly technical descriptions and not presentation of the observations. In addition, only selected time frames are presented in both manuscript and supplement. I would encourage the authors to include a complete timeline of their measurements and certainly move the largest part of this section (3.1) in the methods.

    We have now moved the majority of the text from this section to Section 2.2.1 Compound Identification per recommendation and have adjusted Section 3.1 to more thoroughly describe the observations and include a full timeline of the speciated sesquiterpene and diterpene measurements as follows:

    3.1 Chemical characterization of observed sesquiterpenes

    Thirty sesquiterpene species were observed regularly in the gas phase in SV-TAG chromatograms during the GoAmazon campaign at T3 (Figure 1). Compound names for those compounds positively identified via MS matching and retention index are labelled accordingly in chromatograms and listed with mean concentrations observed during the wet and dry seasons in Table 1. Most sesquiterpene species were observed at mean levels above 100 ppq$_v$, ranging 1-529 ppq$_v$ in the wet season and 1-670 ppq$_v$ in the dry season. While mean observed concentrations differed for some species observed in the wet and dry season, overall summed mean concentrations of sesquiterpenes was similar in both seasons (~4-5 ppq$_v$).

    Complete timelines of speciated sesquiterpens are presented in Figure 2 wet season and Figure 3 dry season. In panel a), six species are presented with hourly time-resolution under

regularly derivatized run conditions. In panel b), occasional runs without derivatization allow for complete speciation of sesquiterpene/diterpene species and to calculate summed concentration of sesquiterpenes and diterpenes as overlaid in panel a). For both seasons, sesquiterpenes exhibit highest concentrations overnight. Note also that a more dynamic range of summed sesquiterpene concentrations were observed during the wet season (spanning across 15 ppt$_v$), whereas in the dry season the range is closer to 8 ppt$_v$. Further, the wet season exhibits the greater chemodiversity of observed sesquiterpenes and terpenes compared to the dry season.

New figures:

[Figure]

**Figure 2: Wet season timeline of sesquiterpene (SQT) and diterpene species for those measured with hourly time resolution with derivatization (shaded colors) a) and those measured multiple times per day at lower time-resolution without derivatization b). Legend entries correspond to compound numbers in Table 1. Total SQT and diterpenes quantified during runs without derivatization are overlaid in black for reference in panel a). Concurrent speciation shown in b).**

[Figure]

**Figure 3: Dry season timeline of sesquiterpene (SQT) and diterpene species for those measured with hourly time resolution with derivatization (shaded colors) a) and those measured multiple times per day at lower time-resolution without derivatization b). Legend entries correspond to compound numbers in Table 1 and colors in b) same as that used in Figure 1. b). Total SQT and diterpenes quantified during runs without derivatization are overlaid in black for reference in panel a). Concurrent speciation shown in b).**

3. While the uncertainties on the reaction rate constants are discussed for sesquiterpenes, the same rate constant as a-pinene has been applied for monoterpenes. Yanez-Serrano et al. (2018) demonstrated a similar chemodiversity of monoterpenes for both wet and dry season inside the Amazon rainforest. Therefore, the uncertainties of monoterpene reactivity (and hence the relative contribution to isoprene and sesquiterpenes) can be minimized. I recommend re-calculation of the O3 reactivity based on the monoterpene speciation from the literature with the respective reaction rates and relative abundance.

The authors appreciate that the referee has brought up this point for discussion. The analysis recommended by the referee was considered in original preparation of the manuscript, but was then simplified to the current version for several reasons which we expand on here. First of all, while the ideal constraint on monoterpenes contribution to $O_3$ reactivity at the measurement site is to have speciated measurements of monoterpenes at T3, these data unfortunately do not exist. To simply assume that the chemodiversity of monoterpenes observed within-canopy presented in Yanez-Serrano et al., (2018) and Jardine et al., (2015) applies to T3 would also introduce uncertainty to this analysis and may even be more erroneous than using the selected rate constant for α-pinene for the following reasons:

a) Mean concentration of total monoterpenes measured within-canopy at several sites around the region are 0.82 ± 0.34 ppbv day 24m height, 0.45 ± 0.13 ppbv night 24 m height for the dry season (Yáñez-Serrano et al., 2018), 0.67 ± 0.3 ppbv for the wet season and 0.47 ± 0.2

ppbv for the dry season (Alves et al., 2016), and 1.3 ppbv averaged for a period spanning the dry-to-wet transition, wet, and wet-to dry transition seasons (Jardine et al., 2015). At T3, mean concentrations of monoterpenes were $0.15 \pm 0.09$ ppbv for the wet season and $0.21 \pm 0.12$ ppbv for the dry season. The comparison of monoterpene concentrations within canopy presented in these literature and that at T3 suggests that there has been significant losses (reactive, deposition, etc.) within the canopy and during transport outside of the canopy.

b) Since the kO3 rate constants for the observed monoterpenes near the source of emission span over two orders of magnitude, this also means that the monoterpene composition at T3 will be dissimilar to that within the canopy. If the within-canopy monoterpene speciation were assumed to be the same at T3 as in the literature, then certainly O3 reactive loss due to reaction with monoterpenes would be more evenly spread across more reactive species (e.g. d-limonene, α-terpinene, cis-β-ocimene) rather than α-pinene, which would make up < 10%, but we know this to be an incorrect approach because of a) above and c) below.

c) If one were to "react" the within canopy-level monoterpenes speciated in Yáñez-Serrano et al., (2018) with even 30 ppbv O3 levels (representative of daytime O3 concentrations during dry season at T3) to achieve the monoterpenes concentrations measured at T3, this would be equivalent to 8 hours of reaction time, which is much longer than the expected transport time (~8 mins) from the nearest surrounding trees (~2.5 km, ~5 m/s wind speeds). For reference, the measurement site, T0a/ATTO, of Yáñez-Serrano et al., (2018) is ~225 km northeast of T3, which would be approximately 12.5 hrs transport time. This mismatch in transport timescales and summed monoterpene concentrations supports that the chemodiversity and concentrations of monoterpenes must vary dramatically in space and time. At T3, the less chemically reactive species will remain. In fact, to render the observed monoterpenes concentration at T3 via reaction with O3 starting with canopy-level monoterpenes speciation, the O3 reactivity associated with monoterpenes left at T3 would be dominated by α-pinene (81%), β-pinene (10%), and d-limonene (5%). No other reported monoterpene species would make up > 1% of the O3 reactivity from monoterpenes. The selection of using the reaction rate constant from α-pinene vs. an assumed monoterpene chemical composition as stated previously would render an "overestimate" of O3 reactive loss due to monoterpenes by a factor of 2.8. Using an average kO3 for the monoterpenes observed at T0a/ATTO by Yáñez-Serrano et al., (2018) would render an even greater overestimate by a factor of 100. An analogous analysis using the monoterpene speciation at T0k measurement site in Jardine et al., (2015), approximately 100 km northeast of T3 with 20 ppbv O3 levels (representative of daytime O3 concentrations during wet season at T3) requires 6 hours of reaction time to achieve T3 monoterpene levels. While O3 reactivity at T3 would be largely attributed to α-pinene (43%), d-limonene (30%), sabinene (18%), and β-pinene (6%), using the α-pinene rate constant only results in an overestimate of monoterpene reactivity by a factor of 1.3. Using an average kO3 for the monoterpene speciation at T0k results in an "overestimate" by a factor of 7.7. Thus, the selection of using α-pinene rate constant is not unreasonable given that much greater error could be introduced assuming that T3 monoterpene speciation is the same as that within the canopy.

In light of the above considerations, we have adjusted the text pg. 10, lines 14-16 to the following:

"Further, the estimate for monoterpene contribution to O3 reactivity assumes that all monoterpenes here have the same rate constant as α-pinene, as monoterpene measurements were not speciated here and it is one of the more dominant (17% and 45% by mass) and longer-lived monoterpenes as observed in upwind forested sites by Jardine et al., (2015) and Yáñez-Serrano et al., (2018). As observed concentrations of monoterpenes within canopy are a factor of 3-4 higher than those observed at T3, it is reasonable to expect that O3 loss due to reaction with monoterpenes at this measurement site will become increasingly dominated by reaction with α-pinene."

General technical comments:

1. Please ensure that the supplementary material is appropriately cited in the main text.

   The cross-references to supplementary material have been revised and appropriately cited.

2. Please ensure that your references conform to the ACP style.

   The references have been revised to conform to the ACP style.

Specific comments:

1. P3L25-28. No need to repeat the measurement challenges as they were already mentioned above.

   The measurement challenges here refer to those associated with measurement of tracers of oxidation of terpenes, some of which also apply to the measurement of the terpenes themselves mentioned above. We prefer to keep this text here to keep the distinction.

2. P4L5. You may keep the definition of IOP but it would be better if you refer to your periods as wet and dry season thereafter.

   We have adjusted singular references to IOP1 and IOP2 and replaced them with wet and dry seasons, respectively.

3. P4L17. There is no need for this last sentence.

   This sentence has been deleted.

4. P8L31. Please rename as "Results and discussion". As mentioned above, I would recommend to include a section over which the observations are described.

   The section has been renamed as suggested and we have included a section for describing the observations according to General Comment 2 above.

5. P8L36. This class of compounds is referred as sesquiterpenoids in Chan et al. (2016).

   We have revised this class to be referred to as "sesquiterpenoids", and these lines have been moved to Section 2.2.1 Compound Identification.

6. P9L24. Please site the "previous literature". P9L34-36. Did you observe such case? Is there a possibility of presenting a case study?

   We refer to the analysis in which we compare with the sesquiterpene concentrations observed by Alves et al. (2016) to those that we have measured. We have changed "previous literature" to cite Alves et al. (2016) specifically.

7. P10L10-14 and L30. It would be interesting if an upper end of sesquiterpene estimated O3 reactivity is presented as well.

   While we agree that this would be interesting, we found original analyses in this regard to be highly speculative to include in this manuscript. For the referee's interest, when we considered an estimate of β-caryophyllene that may have reacted by scaling the observed α-copaene concentration by the ratio of β-caryophyllene: α-copaene in copaiba essential oil (Table S3, Young Living Essential Oil, origin Brazil), we calculated that sesquiterpene reactivity would be higher by a factor of ~4 and would then contribute approximately equally to if not the majority of the reactive O$_3$ loss. Reaction of O$_3$ with isoprene would account for 31%, 33%, monoterpenes for 30%, 31%, and sesquiterpenes for 39%, 36% during the wet, dry seasons respectively. Still, as the referee points out

8. P10L14-16. Please see my general comment.

   This has been addressed in General Comment 4 above.

9. P11L21-22. Nonetheless, this is your practice for monoterpenes. Please discuss a quantitative

   This comment was cut off in the submitted text from the reviewer so we could not read it or address it.

10. Figure 6. The filter tracers are not visible. Maybe the use of log scale would help?

    Thank you for the suggestion, though unfortunately log scale does not help significantly.

References:

Yáñez-Serrano, A. M., Nölscher, A. C., Bourtsoukidis, E., Gomes Alves, E., Ganzeveld, L., Bonn, B., Wolff, S., Sa, M., Yamasoe, M., Williams, J., Andreae, M. O., and Kesselmeier, J.: Monoterpene chemical speciation in a tropical rainforest:variation with season, height, and time of day at the Amazon Tall Tower Observatory (ATTO), Atmos. Chem. Phys., 18, 3403-3418, https://doi.org/10.5194/acp-18-3403-2018, 2018.

Additional changes by authors:

Improvements in calibration for beta-nocaryophyllonic acid using an authentic standard for UHPLC-MS analyses have occurred and data and analyses are updated accordingly in the revised manuscript.

References in authors' response:

Alves, E. G., Jardine, K. J., Tota, J., Jardine, A. B., Yáñez-Serrano, A. M., Karl, T., Tavares, J., Nelson, B., Gu, D., Stavrakou, T., Martin, S. T., Artaxo, P. and Manzi, A.: Seasonality of isoprenoid emissions from a primary rainforest in central Amazonia, Atmos. Chem. Phys., 16, 3903–3925, doi:10.5194/acp-16-3903-2016, 2016.

Isaacman-VanWertz, G., Yee, L. D., Kreisberg, N. M., Wernis, R., Moss, J. A., Hering, S. V., de Sá, S. S., Martin, S. T., Alexander, M. L., Palm, B. B., Hu, W. W., Campuzano-Jost, P., Day, D. A., Jimenez, J. L., Riva, M., Surratt, J. D., Viegas, J., Manzi, A., Edgerton, E. S., Baumann, K., Souza, R., Artaxo, P. and Goldstein, A. H.: Ambient Gas-Particle Partitioning of Tracers for Biogenic Oxidation, Environ. Sci. Technol., 50(18), 9952–9962, doi:10.1021/acs.est.6b01674, 2016.

Isaacman, G. A., Kreisberg, N. M., Worton, D. R., Hering, S. V and Goldstein, A. H.: A versatile and reproducible automatic injection system for liquid standard introduction: Application to in-situ calibration, Atmos. Meas. Tech., 4(9), 1937–1942, doi:10.5194/amt-4-1937-2011, 2011.

Jardine, A. B., Jardine, K. J., Fuentes, J. D., Martin, S. T., Martins, G., Durgante, F., Carneiro, V., Higuchi, N., Manzi, A. O. and Chambers, J. Q.: Highly reactive light-dependent monoterpenes in the Amazon, Geophys. Res. Lett., 42(5), 1576–1583, doi:10.1002/2014GL062573, 2015.

Stein, S. E.: Estimating probabilities of correct identification from results of mass spectral library searches, J. Am. Soc. Mass Spectrom., 5(4), 316–323, doi:10.1016/1044-0305(94)85022-4, 1994.

Yáñez-Serrano, A. M., Nölscher, C., Bourtsoukidis, E., Gomes Alves, E., Ganzeveld, L., Bonn, B., Wolff, S., Sa, M., Yamasoe, M., Williams, J., Andreae, M. O. and Kesselmeier, J.: Monoterpene chemical speciation in a tropical rainforest: variation with season, height, and time of day at the Amazon Tall Tower Observatory (ATTO), Atmos. Chem. Phys, 185194, 3403–3418, doi:10.5194/acp-18-3403-2018, 2018.